# Facing the COVID-19 pandemic inside maternities in Brazil: A mixed-method study within the REBRACO initiative

Maria Laura Costa[1]*, Renato T. Souza[1]☯, Rodolfo C. Pacagnella[1], Silvana F. Bento[1], Carolina C. Ribeiro-do-Valle[1], Adriana G. Luz[1], Giuliane J. Lajos[1], Guilherme M. Nobrega[1], Thayna B. Griggio[1], Charles M. Charles[1], Ricardo P. Tedesco[2], Karayna G. Fernandes[2], Sérgio H. A. Martins-Costa[3], Frederico J. A. Peret[4], Francisco E. Feitosa[5], Rosiane Mattar[6], Edson V. Cunha Filho[7], Janete Vetorazzi[3,7], Samira M. Haddad[8], Carla B. Andreucci[9], José P. Guida[10], Mário D. Correa Junior[11], Marcos A. B. Dias[12], Leandro G. Oliveira[13], Elias F. Melo Junior[14], Carlos A. S. Menezes[15], Marília G. Q. Luz[16], Jose G. Cecatti[1], for the REBRACO Study Group¶

1 Department of Obstetrics and Gynecology, University of Campinas, Campinas/SP, Brazil, 2 Faculdade de Medicina de Jundiaí HU/FMJ, Jundiaí/SP, Brazil, 3 Hospital das Clínicas de Porto Alegre, Porto Alegre/RS, Brazil, 4 Maternidade UNIMED UNIMED/BH, Belo Horizonte/MG, Brazil, 5 Universidade Federal do Ceará MEAC/UFC, Fortaleza/CE, Brazil, 6 Universidade Federal de São Paulo UNIFESP/EPM, São Paulo/SP, Brazil, 7 Hospital Moinhos de Vento-HMV, Porto Alegre/R, Brazil, 8 Hospital Regional Jorge Rossmann Instituto Sócrates Guanaes, Itanhaém/SP, Brazil, 9 Universidade Federal de São Carlos/UFSCAR, São Carlos/SP, Brazil, 10 Hospital Estadual Sumaré HES, Sumaré/SP, Brazil, 11 Universidade Federal de Minas Gerais HC/UFMG, Belo Horizonte/MG, Brazil, 12 Instituto Fernandes Figueira IFF/Fiocruz, Rio de Janeiro/RJ, Brazil, 13 Faculdade de Medicina da Universidade Estadual de São Paulo, Botucatu/SP, Brazil, 14 Universidade Federal de Pernambuco HC/UFPE, Recife/PE, Brazil, 15 Maternidade Climério de Oliveira MCO-UFBA, Salvador/BA, Brazil, 16 Santa Casa de Misericórdia do Pará, Belém/PA, Brazil

☯ These authors contributed equally to this work.
¶ Membership of the REBRACO research team is provided in the Acknowledgments.
* mlaura@unicamp.br

**Data Availability Statement:** All relevant data are within the paper and its Supporting Information files.

## Abstract

### Introduction

COVID-19 pandemic posed major challenges in obstetric health care services. Preparedness, development, and implementation of new protocols were part of the needed response. This study aims to describe the strategies implemented and the perspectives of health managers on the challenges to face the pandemic in 16 different maternity hospitals that comprise a multicenter study in Brazil, called REBRACO (Brazilian network of COVID-19 during pregnancy).

### Methods

Mixed-method study, with quantitative and qualitative approaches. Quantitative data on the infrastructure of the units, maternal and perinatal health indicators, modifications on staff and human resources, from January to July/2020. Also, information on total number of cases, and availability for COVID-19 testing. A qualitative study by purposeful and saturation sampling was undertaken with healthcare managers, to understand perspectives on local challenges in facing the pandemic.

**Funding:** The funders had no role in study design, data collection and analysis, decision to publish, or preparation of the manuscript. The funding source of support during this study was FAEPEX-Unicamp (Fundo de Apoio ao Ensino, à Pesquisa e à Extensão) under grant number 2300/20.

**Competing interests:** The authors have declared that no competing interests exist.

## Results

Most maternities early implemented their contingency plan. REBRACO centers reported 338 confirmed COVID-19 cases among pregnant and post-partum women up to July 2020. There were 29 maternal deaths and 15 (51.8%) attributed to COVID-19. All maternities performed relocation of beds designated to labor ward, most (75%) acquired mechanical ventilators, only the minority (25%) installed new negative air pressure rooms. Considering human resources, around 40% hired extra health professionals and increased weekly workload and the majority (68.7%) also suspended annual leaves. Only one center implemented universal screening for childbirth and 6 (37.5%) implemented COVID-19 testing for all suspected cases, while around 60% of the centers only tested moderate/severe cases with hospital admission. Qualitative results showed that main challenges experienced were related to the fear of the virus, concerns about reliability of evidence and lack of resources, with a clear need for mental health support among health professionals.

## Conclusion

Study findings suggest that maternities of the REBRACO initiative underwent major changes in facing the pandemic, with limitations on testing, difficulties in infrastructure and human resources. Leadership, continuous training, implementation of evidence-based protocols and collaborative initiatives are key to transpose the fear of the virus and ascertain adequate healthcare inside maternities, especially in low and middle-income settings. Policy makers need to address the specificities in considering reproductive health and childbirth during the COVID-19 pandemic and prioritize research and timely testing availability.

## Introduction

Coronavirus disease 2019 (COVID-19) has threatened the world, since March 2020, when the pandemic was officially declared by the WHO (World Health Organization) [1]. Some countries were especially affected and faced individual challenges towards viral dissemination. Brazil, was certainly one of those, recognized as a pandemic hotspot with more than 4.1 million people infected and over 126,000 deaths, in the first 6 months of the pandemic [2]. Brazil is an upper-middle-income country of continental size, with great social and economic disparities among different regions, and major political crisis. These characteristics enhance the challenges in facing a pandemic [3, 4].

The impact of Covid-19 during pregnancy is still under investigation, with many unanswered questions, involving risk of vertical transmission, reinfection, possible increased risk of severe complications, long-term sequels and also concerns involving treatment, fetal assessment, route and timing of delivery [5–9]. Just as importantly, there are great concerns regarding the indirect effects of the pandemic on the access and availability of healthcare services, especially for women's health. Differently from other clinical conditions or elective procedures, obstetric care cannot be postponed, and childbirth cannot be rescheduled, and low-resourced settings have been facing a great challenge to ascertain proper care.

Most statistics focus on the number of infected people, hospital admissions and deaths, but little is considered on the efforts to prepare each institution and their health professionals for healthcare during the pandemic. This challenge is even greater in maternity hospitals, many of

them with a limited number of beds, especially intensive care unit beds and adequate clinical support. In the earliest stages of the pandemic, the Brazilian Network for Studies in Reproductive and Perinatal Health [10] established a collaborative multicenter approach in Brazil, called REBRACO (Brazilian network of COVID-19 during pregnancy, in Portuguese: *REde BRAsileira em estudos do COVID-19 em Obstetrícia*). Overall, the REBRACO initiative aimed at evaluating the clinical, epidemiological and laboratory aspects related to SARS-CoV-2 infection during pregnancy, besides a qualitative assessment of women and professionals experiencing such a situation, to identify maternal and perinatal outcomes and collect relevant information to provide quick responses and proper organization of health services to confront the COVID-19 pandemic [11].

As part of the main study, we aim to describe the strategies implemented and the perspectives of health managers on the challenges to face the pandemic in 16 different maternity hospitals of the REBRACO initiative around Brazil, using quantitative and qualitative approaches.

## Materials and methods

A national multicenter study (REBRACO) was implemented involving 16 maternity hospitals in Brazil (Fig 1), with data collection considering information retrieved from January/2020 to July/2020. Centers were invited to participate in the study considering their previous experience in other studies in the network. In each participating center, a local PI was identified as responsible for locally supervising the study, plus one or two local coordinators responsible for identifying cases, filling the electronic forms with pertinent information, and performing other activities and procedures of the study.

Ethical approval for the main study and in each participating center was obtained (Letter of Approval numbers 4.047.168, 4.179.679, and 4.083.988, from the coordinating center´s Institutional Review Board from the University of Campinas, the IRB of the School of Medicine of Jundiai, IRB of the Clinics Hospital of Porto Alegre, the IRB of the Unimed Maternity of Belo Horizonte, the IRB of MEAC from the Federal University of Ceara, the IRB of the Federal University of Sao Paulo, IRB of Moinhos de Vento Hospital from Porto Alegre, IRB of the Jorge Rossmann Regional Hospital from Itanhaem, IRB of the Federal University of Sao Carlos, IRB of the State Hospital of Sumare, IR of the Feral University of Minas Gerais, IRB of the Fernandes Figueira Institute from Fiocruz in Rio de Janeiro, IRB of the School of Medicine from the State University of Sao Paulo in Botucatu, IRB of the Federal University of Pernambuco, IRB of the Climerio de Oliveira Maternity from the Federal University of Bahia in Salvador, and the IRB of the Santa Casa de Misericordia do Pará from Belem.

### Quantitative component

For the quantitative component, we collected data on the infrastructure/equipment of the units, maternal and perinatal health indicators, characteristics of service provision and modifications on staff and human resources. All information was retrieved through an electronic form developed by the coordinating center and sent by e-mail to the Principal Investigator of each center (Health Manager, responsible for the Obstetrical Emergency Action Committee—EAC—in each institution). The data collection variables and form are provided as S1 and S2 Datas and includes data on: characteristics of the maternities, their response program, their training program and some indicators related to the COVID-19 pandemic such as the number of cases, number of maternal deaths, maternal mortality ratio due to COVID-19 in the period, the time interval between the implementation of the response program and the first suspected COVID-19 cases in each center and the characteristics of the tests available in the obstetric units.

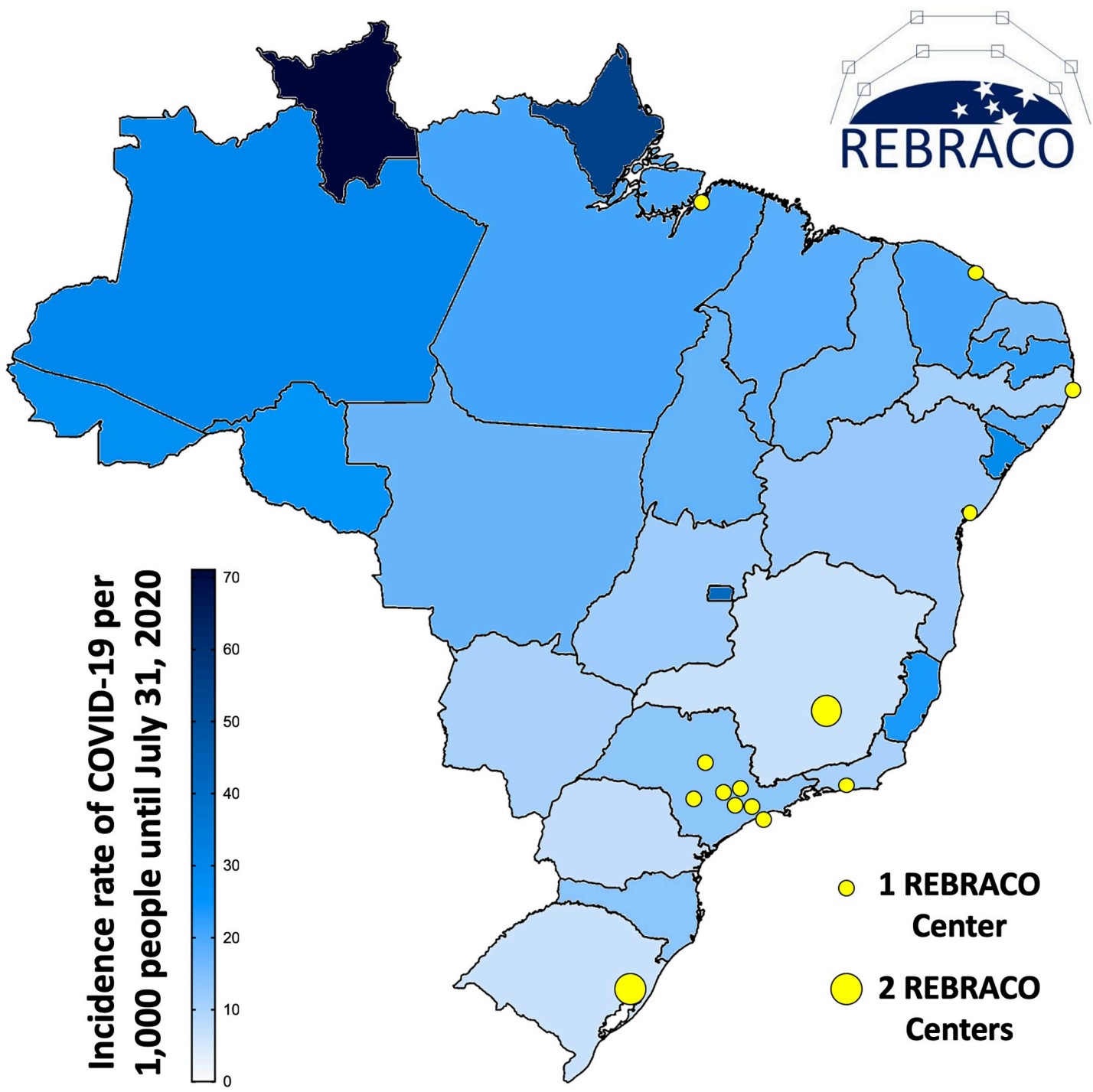

**Fig 1. Brazil political-geographic map showing its federative units (states), and the location of the participating centers of REBRACO.** States colored according to the incidence rate of confirmed cases of COVID-19 per thousand people until July 31, 2020 [2, 12, 13]. Fig 1 was adapted from a map contained in the Brazilian public domain database–Portal Domínio Público (http://www.dominiopublico.gov.br).

All suspected cases of COVID-19 in the considered maternities were assessed for informed consent and their data was only included after such consent, with identity kept confidential. Considering biosafety reasons, the study has authorization, by the Institutional Review Board,

for both- written informed consent and consent by phone to use information from medical records and sample collection. Mostly suspected cases with hospital admissions have written informed consent, cases that were evaluated and tested in the emergency room, with no needed admission were later consented by phone. The researcher read and explained the consent form and assured that women could withdraw from the study without any interference with planned medical care.

The data sharing information on the questionnaire of readiness towards the pandemic and overall numbers of each institution was approved by the coordinating center and locally in each participating center. Data was stored in a specific database created for this study and protected by password in a de-identified way.

### Qualitative component

We also performed a qualitative component, considering key obstetricians involved in the decision making of each institution, to listen to their perspectives on the challenges to face the pandemic in their setting. These participants were identified by local PIs (principal investigators). A qualitative study by purposeful sampling and saturation sampling was undertaken after individual oral consent. All semi-structured interviews were performed by a skilled social researcher with experience in the field and were recorded on audio. For this approach, the Institutional Review Board (IRB) approved the use of oral consent. Maternities considered are in different regions of the country and all interviews were performed by the same researcher in the coordinating center, by telephone, with specific software for further transcription and analysis ("ReShape"), protected by password in a de-identified manner. All interviews were recorded on audio after consent from the participants, using the Microsoft Skype platform.

The semi-structured interview script for health managers, used on the qualitative study is provided as S3 Data. Subsequently, the recordings were verbatim transcribed, and the text obtained checked with the recording. The texts were further inserted in the NVivo® computer program to perform the analysis. Thematic analysis was finally carried out [14] after the familiarization with the collected data. Then we identified the main themes after generating the initial codes. The themes were reviewed and, finally, they were defined and named according to the main challenges identified in the interviewees' narratives, with their corresponding experiences, conflicts and answers/actions. The analysis was divided into four categories: 1. Preparedness and implementation of actions to fight the pandemic; 2. Taking care of women: information and solutions; 3. Assisting health professionals; 4. The burden of leadership.

In the REBRACO initiative, there are no planned interventions in each included centers. All the research team, including local-PIs, health professionals involved in the local Emergency Action Committees, and other collaborators (Ob&Gyn residents, consultants, etc.) from all the included centers, have been invited to participate in weekly virtual meetings, sharing information regarding the organization of the different health services, the barriers and facilitators in the implementation of healthcare and training of health professionals in each center and discussions on current findings on maternal and perinatal outcomes during pregnancy and postpartum of COVID-19 infected women. These meetings have inspired the current analysis.

### Results

Table 1 shows the main characteristics of included maternities in the REBRACO study (n = 16), participating members of the local Emergency Action Committees (EAC) and information on COVID-19 infection and some healthcare indicators. The majority of the centers are public maternities (n = 12) and develops teaching activities (n = 15), with more than 300 deliveries/month, and mostly playing the role of local referral centers for managing COVID-

**Table 1. Characteristics of included maternities in the REBRACO study (n = 16), participating members of the local Emergency Action Committees (EAC) and information on COVID-19 infection and some healthcare indicators.**

| | Number of maternities (%) |
|---|---|
| **Characteristics** | |
| Public Maternities | 12 (75%) |
| Centers with teaching activities | 15 (93.8%) |
| Deliveries per month | |
| <300 | 7 (43.7%) |
| ≥ 300 and <500 | 8 (50.0%) |
| ≥ 500 | 1 (6.3%) |
| Referral obstetric unit for COVID-19 cases | 12 (75%) |
| Access to laboratory diagnostic tests | 16 (100%) |
| Possibility to test any suspected case of COVID-19 | 6 (37.5%) |
| **Members of the local Emergency Action Committees (EAC)** | |
| Obstetrics and Gynecology specialist | 12 (75%) |
| Member of the Infection Prevention and Control Committee | 12 (75%) |
| Nurse | 12 (75%) |
| General practitioner or internal medicine | 11 (73%) |
| Intensive care specialist | 11 (68.7%) |
| Administrative assistant | 11 (68.7%) |
| Physiotherapist | 5 (31.2%) |
| **COVID-19 infection and Healthcare indicators (Feb-Jul/2020)** | |
| Total confirmed cases among pregnancy/postpartum (n)* | 338 |
| COVID-19 incidence cases** | 14,9 / 1,000 LB |
| Maternal Death | 29 |
| MD due to COVID-19 (n;%) | 15 (51.8%) |
| Maternal Mortality Ratio (MMR) | 127.8 / 100,000 LB |

* Most likely underestimated preliminary result- all centers are auditing their official information to clarify these numbers- mostly these are cases admitted to the hospital. Many centers only test symptomatic cases that need hospital admission.

**There were 22,690 live births in the participating centers in the period (Feb-Jul/2020). LB: live births; MD: Maternal deaths.

19 suspected and confirmed cases (n = 12). These maternities respond for around 4,000 deliveries per month in the country, with 338 confirmed COVID-19 cases among pregnant and post-partum women up to July 2020. It represented an incidence of 14.9 cases for 1,000 live births. There were overall 29 maternal deaths (distributed in only 7 of the 16 centers) and 15 (51.8%) attributed to COVID-19 (in 6 centers), with a Maternal mortality rate (MMR) of 127.8/100.000 LB.

All maternities established a local Covid-19 Emergency Action Committee (EAC) and implemented a local protocol for contingency and management of COVID-19 in pregnancy. The EAC was usually comprised of a general practitioner or internal medicine (73% of the centers, n = 11), Obstetrics and Gynecology specialist (75%, n = 12), intensive care specialist (68.7%, n = 11), nurse (75%, n = 12), member of the Infection Prevention and Control Committee (75%, n = 12), administrative assistant (68.7%, n = 11), and, less often, of a physiotherapist (31.2%, n = 5) (Table 1). All the 16 maternities had access to diagnostic laboratory testing. However, only 6 maternities (37.5%) had resources to test any suspected case of COVID-19. Fig 2 shows the characteristics of the tests performed in the REBRACO centers. In half of the

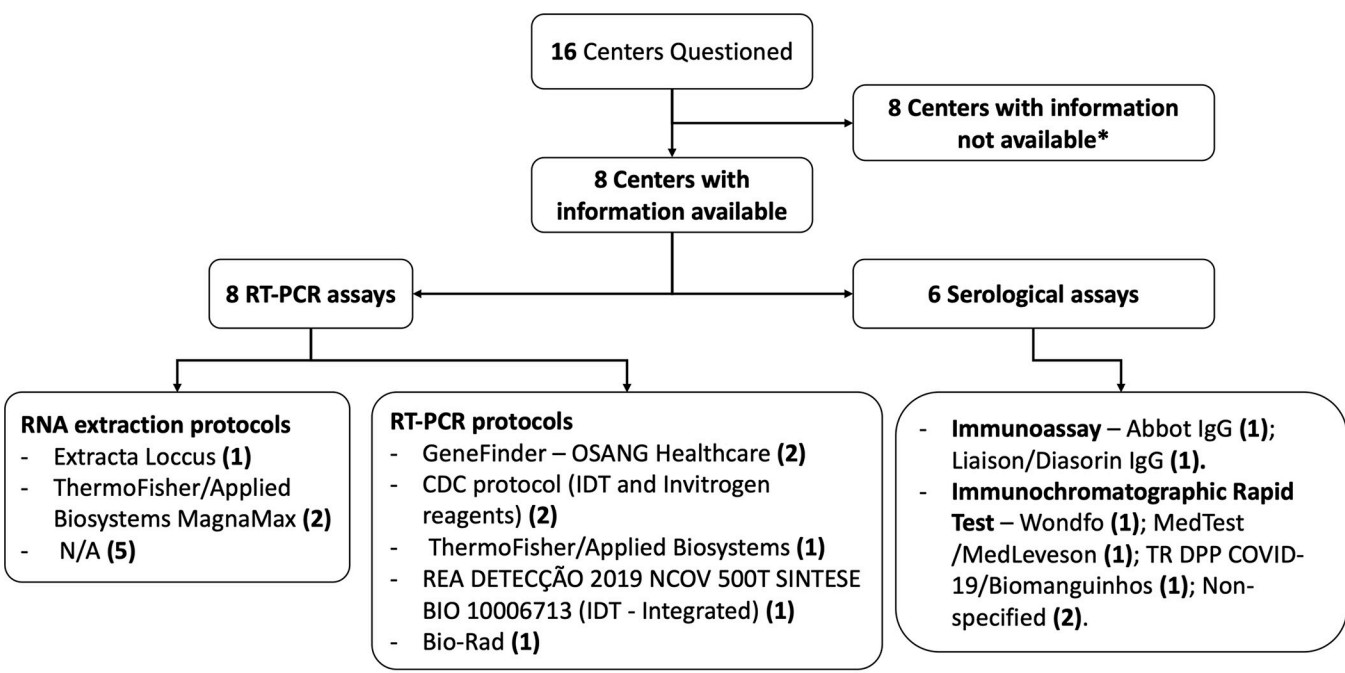

* Tests analyzed at another institution – protocol not available

**Fig 2. Characteristics of the tests for COVID-19 available in the REBRACO centers.**

centers (n = 8), the tests were not performed in the facility and the characteristics were not available. The characteristic of the RT-PCR and serological tests available were heterogeneous across centers. The majority (n = 10) of the units have tested only suspected cases of COVID-19 that presented with severe symptoms requiring inpatient medical support. Only one maternity implemented universal screening to all women admitted for childbirth (implemented in late June 2020). On average, the test results were available in 4 days. Nine maternities had test results in 2 days or less, while other 2 over 7 days.

Considering the progression of the pandemic in Brazil, all maternities developed and launched the local C&M protocol, and started training professionals/staff at early stages of the pandemic. One maternity implemented and trained the team in January, one in February, ten maternities in March and four maternities in April 2020 (Fig 3). The mean time between the implementation of the C&M protocol and the identification of the first confirmed Covid-19 case was 71 days. Only one of the maternities registered the first case of COVID-19 before implementing the contingency and management (C&M) protocols. The moment when the first case was identified varied among the maternities (Fig 3): two maternities registered the first case in early March and the majority of the maternities identified the first case in April and May (n = 11).

An isolated and especially COVID-19 designated private area in the Emergency Room and the medical ward were available in only 12 (75%) and 5 (30%) maternities, respectively. The availability of isolated beds varied according to the size of the maternity measured by the number of deliveries. There was one isolated bed reserved for COVID-19 for every 2.6 deliveries/month in maternities with ≥300 deliveries/month whereas there was one isolated bed for every 15.3 deliveries/month in maternities with <300 deliveries/month. Almost all maternities

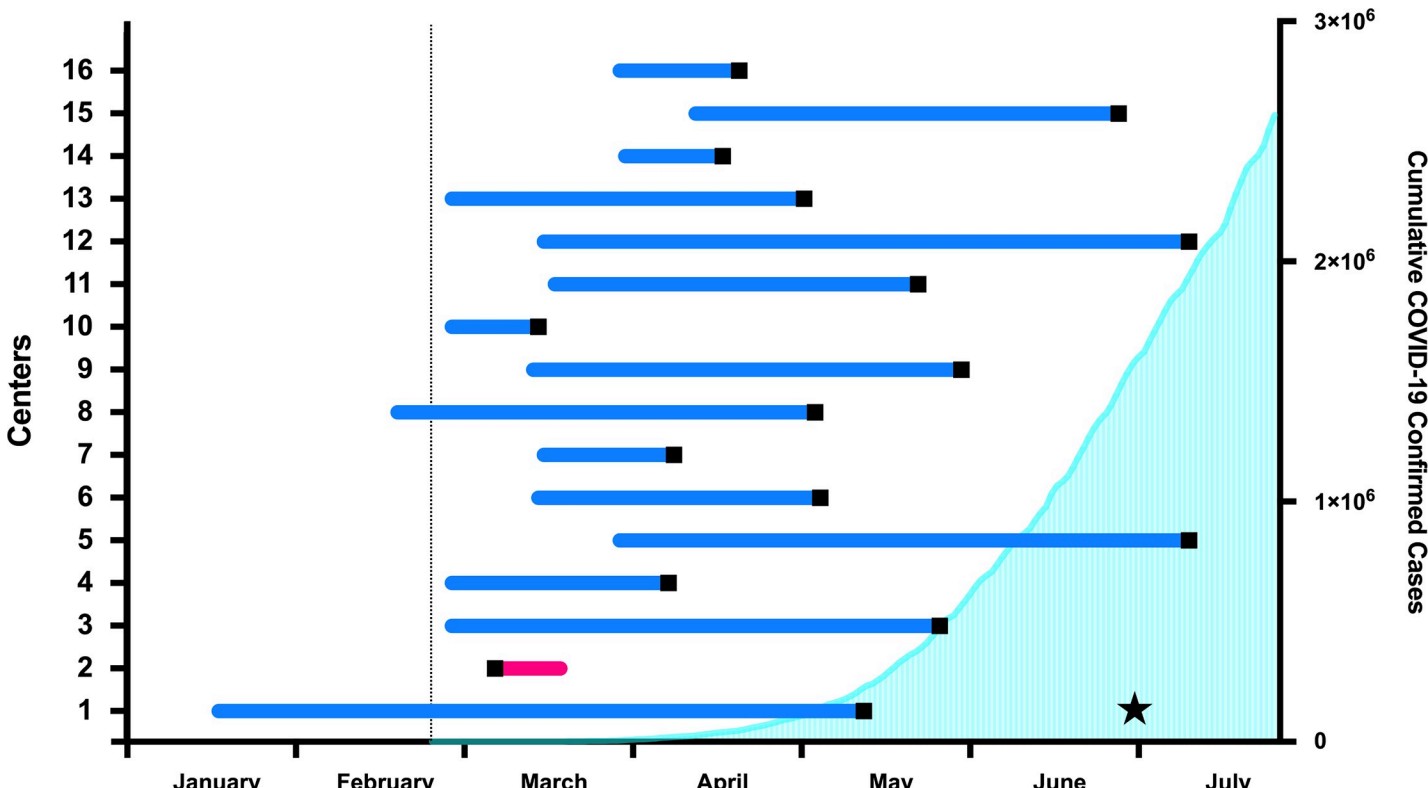

**Fig 3. COVID-19 pandemic progression in Brazil concomitant with the implementation of the contingency and management (C&M) protocols in each REBRACO center.** Horizontal bars represent the interval from C&M implementation and the first confirmed COVID-19 case in each center; blue indicates C&M implementation prior to the first case reported and magenta represents indicates C&M implementation after the first case. The black square in each bar indicates the first case recorded at each center. The star represents the initial date of COVID-19 universal screening at center 1. Turquoise curve represents the cumulative number of confirmed cases in Brazil until July 31, 2020. Vertical dashed line marks the first confirmed case registered in Brazil [2, 13].

had an ICU within their hospital (n = 15), and one third had a maternal-fetal ICU in the hospital. Half of the maternities have implemented an isolated and specially designated COVID-19 ICU.

**Table 2. Changes on infrastructure and human resources due to COVID-19 pandemic in the REBRACO maternities (n = 16).**

| | Number of maternities (%) |
|---|---|
| **Infrastructure/Equipment** | |
| Resizing the Emergency Room area due to new healthcare flow processes | 11 (68.7%) |
| Relocation of beds designated to labor ward, medical wards and ICU | 16 (100%) |
| Installation of negative air pressure room (temporary or definitive solution) | 4 (25.0%) |
| Acquisition of (new) mechanical ventilators | 12 (75.0%) |
| **Staff/Human resources** | |
| Hiring health professionals | 6 (37.5%) |
| Increases in weekly workload | 6 (37.5%) |
| Suspension of annual leaves | 11 (68.7%) |
| Redeploying health professionals at higher risk for severe infection avoiding higher risk sites* | 16 (100%) |

* Redeploying health professionals at higher risk for severe infection avoiding higher risk sites was recommended by the Brazilian Ministry of Health [15].

**Table 3. Training health professionals in the REBRACO maternities (n = 16).**

| Topic | Maternities that performed the training | *In loco*[*] | Written protocol[*] | Webinars (videos)[*] |
|---|---|---|---|---|
| Screening/identifying suspected cases | 14 (87.5%) | 9 (64.2%) | 14 (100%) | 6 (42.8%) |
| Putting on and removing PPE | 15 (93.7%) | 13 (86.6%) | 13 (86.6%) | 11 (73.3%) |
| Assistance of severe acute respiratory syndrome | 11 (68.7%) | 10 (90.1%) | 11 (100%) | 7 (63.6%) |
| Proceeding intubation and cardiopulmonary resuscitation | 11 (68.7%) | 11 (100%) | 9 (81.8%) | 8 (72.7%) |

[*] The proportion (%) is provided according to the total number of maternities that performed the training of the respective topic. PPE, personal protective equipment.

Table 2 shows the modifications and adaptations on infrastructure and human resource policies performed by the maternities during the COVID-19 pandemic. The actions performed included relocation of the designated labor ward, inpatient ward and ICU beds, acquisition of additional mechanical ventilators, and restructuring of the Emergency Room area due to new healthcare flow processes. Redeploying health professionals at higher risk for severe infection away from the highest risk sites and suspension of annual leaves were the most common changes in the staff, followed by hiring more professionals and increasing their weekly workload.

Table 3 shows which topics were addressed in the training of the health professionals and how the training sessions were performed. Training on how to proceed orotracheal intubation and cardiopulmonary resuscitation and management of severe acute respiratory syndrome was performed in 11 maternities. Overall, different methods were employed to perform staff training (in loco, written protocol and/or webinars).

## Results of the qualitative component

All health managers were medical doctors specialized in Obstetrics and Gynecology, aged under 59 years old, and had a PhD degree; half of the participants were women and four of the six were responsible for the Obstetric unit for more than 2 years. The results were divided according to the four analyses categories: 1. Preparedness and implementation of actions to fight the pandemic; 2. Taking care of women: information and solutions; 3. Assisting health professionals; 4. The burden of leadership.

1. Preparedness and implementation of actions to fight the pandemic

During the COVID-19 pandemic, there was a growing need for developing and implementing new protocols and health care strategies. The experiences related to these processes included a) developing protocols: concerns regarding the source and reliability of the evidence; b) fear of the virus; c) lack of resources: frontline workforce, PPE and lab tests. Fig 4 summarizes the main conflicts and response actions described by the health managers on this topic.

a. Developing protocols: Concerns regarding the source and reliability of evidence

Excessive information, not always reliable and conflicting recommendations on COVID-19, by different agencies caused insecurity to professionals and raised the need for constant review and update of protocols. Health managers were responsible to review protocols every day or twice a week. As an example, one health manager shared that had to update the institutional protocol 18 times (so far). Growing evidence on a previously unknown disease makes it a challenge to warrant adequate implementation by the health professionals, requiring frequent training and individual and group meetings.

The need for developing protocols and preparing health professionals for fighting the pandemic was not considered a consensus among all the hospital managers and policymakers.

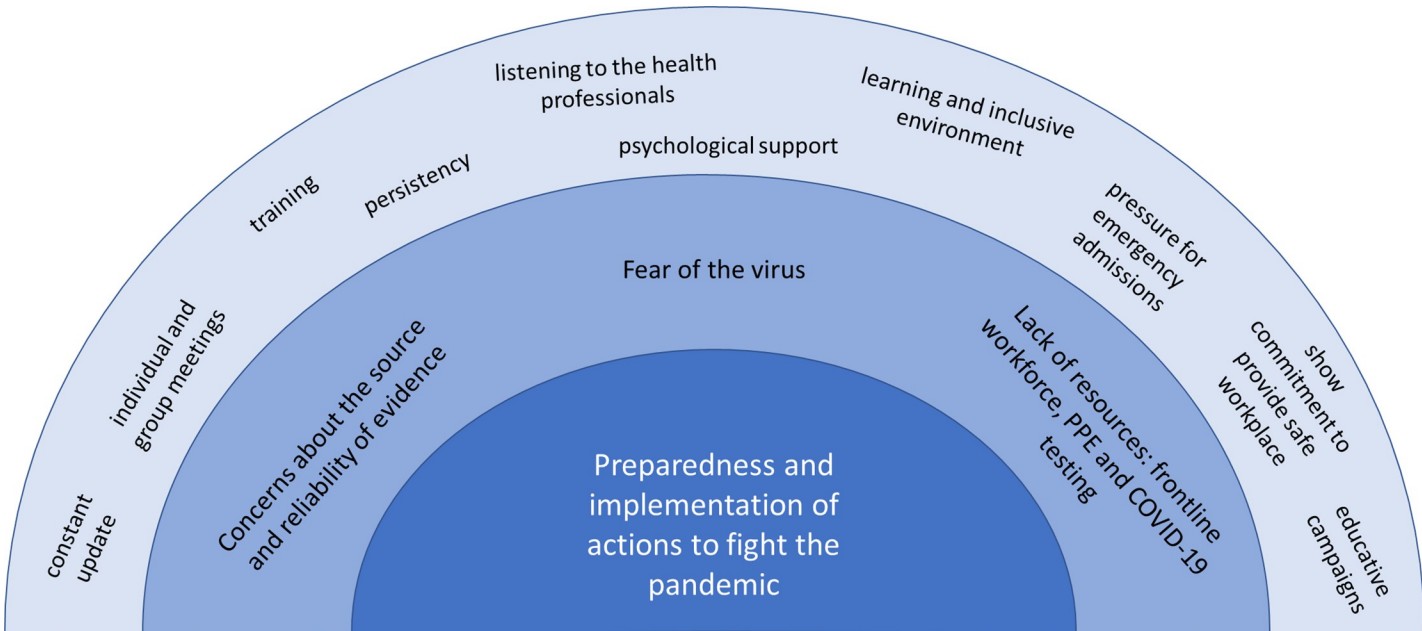

**Fig 4. Framework of the conflicts, experiences and response actions on the preparedness and implementations of actions to fight the pandemic according to the health managers.** The middle layer shows the conflicts and the outer layer demonstrates the main actions performed by the health managers and institutions to tackle the main conflicts.

According to some health managers, the underestimation of the scale of COVID-19 pandemic and the need for preparedness has established a two-front battle in some of the units. Rapidly, the persistency of health managers in showing evidence-based policies required to fight the pandemic, the reaction from health professionals to the pandemic, and the panorama of what was happening in other countries were crucial to convince them of the need for an early response plan.

b.  Fear of the virus

As soon as the first response actions were taken into place, there was a growing fear of providing care to people with suspected or confirmed COVID-19. The uncertainty over the severity and the magnitude of the outbreak in Brazil were the main triggers of fear. Then, the news showing the spread of the virus in other countries and the collapse of their health systems and the great number of deaths that followed. Finally, the fact that health professionals and their families were at higher risk of infection. The fear manifested more strongly when the first patients with suspected COVID-19 started seeking care. The professionals were very scared, cried, and created strategies to avoid working in the frontline (such as missing shifts at COVID-19 ward/frontline). Some medical doctors asked not to provide care to COVID-19 patients and nurses refused to have contact with these patients. Other professionals refused bedside visits, delayed or refused to deliver meals or even to collect samples for laboratory tests. The fear and its consequences resulted in harsh discussions among these professionals and those who were willing to support the patients (they were overloaded).

The fear of the unknown was followed by the fear of being infected and the fear of dying. The limited resources (shortage of beds, ICU, PPE, etc.), an increasing number of deaths, and increase of the workload because of the quarantine of professionals contaminated turned the hospital into a very stressful and toxic work environment. To overcome this challenging scenario, health managers promoted multiple strategies such as small-group discussions on

coping with *COVID*-19 pandemic and on the role of the health professionals, providing mental health services to improve identification and support of related disorders. The role of health care leaders was extremely important to enable all teams to understand how important and critical the role of health professionals is at such difficult times. All these strategies helped to prevent losing the frontline workforce and to create a learning and inclusive environment.

> "*Early during the outbreak, we had a general feeling of fear that was initially resolved. However, when there was no bed available in the city and we started to see people dying near us, there was a big wave of fear again. The [institution] had a mass leave, people who had not given up working began to give up working*" (Health Manager 2)

> "*The big challenge is to demystify; we were afraid of attitudes like doctors who overload the other with their work to avoid contamination. Nurses who avoided contact. . . We did have pleasant surprises, people who also worked very calmly. This is a continuous challenge, as we see medical professionals and mainly nursing professionals talking about getting contaminated. . . We have a lot to do with the safety. We try to work with a high level of safety, gold-standards safety procedures and PPE. But this feeling of insecurity still permeates with fear. People, especially when they see colleagues on duty. . . friends on duty being contaminated*" (Health Manager 4)

c. Lack of resources: frontline workforce, PPE and SARS-CoV-2 testing

A great number of health professionals have got temporary leave due to suspected or confirmed COVID-19 or due to mental health disorders. Those comprising the high-risk group for severe illness had been redeployed to different roles. In addition to the lack of health workers in the frontline, some of the obstetric units have experienced a shortage of PPE (gowns/aprons, masks, gloves, etc.) and tests for COVID-19. Health managers became very worried with the lack of availability of PPE and tests caused by the sharpest increase in the price and delayed delivery of such supplies, contributing to widespread stress, fear and emotional distress.

Health managers promoted fundraising strategies for PPE-related expenses, including donations mostly from the health professionals themselves and the private sector companies. Also, campaigns addressing the rational use of PPE and control in deployment were alternatives employed by the health managers to better use the limited resources. Finally, there was pressure for emergency admissions. These efforts were important to demonstrate the commitment to providing a safe workplace.

> "*We created rules. . . we are going to restrict the waterproof aprons to this procedure in this sector, we are going to restrict the use of a mask; the mask will have to be used for a given period of time. We increased the time of wearing the masks, we needed to assure availability to [non-Covid] companions as well. I needed to assure masks to the patients. So, we made some masks ourselves. We also made aprons for some sectors; we needed to be creative. Then, we received a lot of donations too. [Donation] Mainly of the face shield, about eight hundred; and fabric to make aprons. . . Mask was very difficult; we did not receive [donation] masks. . . no, we needed to fabricate masks. . . we received fabric to make the mask*"(Health Manager 2)

2. Taking care of women: information and solutions

Health managers implemented several educational actions in the units, aiming at protecting pregnant women and their families. These strategies included 1) banners, flyers and folders in the maternity, promoting interviews in radio stations and publishing specific content in the

maternity social media summarizing what pregnant women should know about COVID-19; 2) telehealth solutions for follow-up of women with suspected or confirmed Covid-19 or for guiding women caring of a newborn (breastfeeding, precautions, bonding, etc.); 3) allocating a multidisciplinary team to provide telehealth guidance (nurse, doctor, social service, etc.).

*"We had to create a new program. . . for all who were interested to participate. . . and the adherence was pretty high. . ."* (Health Manager 5)

3. Assisting health professionals

All institutions already had psychological services for health professionals, but now there was an increasing need for facilitating access to such services.

Small group discussions aiming at sharing the same collective challenges helped to increase resiliency, bring back the motivation and show the importance of recognizing their duty and role in the fight against the pandemic. Also, it assisted the identification of those who needed specialized psychological support. Devoting extra attention to the health professionals during this period was positive, providing reassurance at their workplace.

*". . .I think that the health professional realized that they can count on us. . . That they can receive proper care if needed where they work. . . we and the team are closer now, because the disease can affect anyone. I think that this shared experience helped a lot."* (Health Manager 1)

*". . . talking with people was really worthy. . . it was really helpful to reduce their and our anguish and anxiety".* (Health Manager 2)

4. The role and burden of leadership

All health managers felt stressed during the development and implementation of the local C&M protocols. The workload increased, with a great number of virtual meetings and sleep deprivation. While in the professional field they dealt with additional roles in their job position to fight the pandemic. In their personal life, they coped with the loss of colleagues and family members to COVID-19. They needed to extend the time dedicated to working, 24/7 available in some of the periods during the pandemic.

Although challenging and stressful, health managers overall enjoyed being responsible for the job. The commitment of the staff and the opportunity of making the difference were the main motivators.

The experience they had at work has positively affected their life because it was an opportunity to strengthen family bonds. Receiving support and understanding from the family were fundamental to let them cope and deal with multiple tasks during the pandemic, although the routine with the family changed due to the precautions to prevent infection and also because of the lack of time dedicated to partners, children and to focus on exercises or hobbies. Other indirect positive effects were having met other professionals who work on the field and research opportunities. They felt proud for the strong relationship between the members of the C&M group and health professionals, generating a "feeling of brotherhood".

Most of the health managers mentioned that they felt challenged because, in fact, "the feeling of great responsibility" or "as if we were going to war". There was a mixture of feelings and challenges such as being overwhelmed but having support from the families and the staff. Some of the health managers themselves had suspected, but not confirmed, cases of COVID-19, which raised more concerns regarding the safety of their family.

There are relevant personal lessons from the response to the pandemic: to keep a constant dialogue with the health professionals; greater interaction between professionals from different maternity sectors, keeping an inclusive and learning atmosphere, the importance of safety procedures (importance of hand hygiene and use of PPE); improved patient orientation after discharge.

*"I often felt challenged. . . feeling challenged every day. This basically didn't bring me a bad feeling all the time, didn't bring me anxiety, but a feeling of contentment whenever we managed to finish a task."* (Health Manager 2)

*"The biggest challenge was the leadership, the task job of gathering people together and keeping alive the purpose of collaboration that already existed without creating any rupture in the new scenario. That was not easy."* (Manager 4)

## Discussion

The 16 maternities of the REBRACO initiative were mostly university hospitals, which underwent major changes and adaptations in facing the first few months of the pandemic, resizing emergency rooms, relocating beds and adding extra ICU units, with acquisition of ventilators. On human resources, most centers had to employ increased weekly workloads, deal with medical leave due to COVID infections, and hire extra health professionals. Listening to health managers, main concerns included fear of the virus and lack of resources. There was a clear limitation in the capacity of testing during the considered study period, less than half of the centers had resources to test any suspected case of COVID-19.

The challenge to translate evidence into practice during a pandemic, to update clinical protocols, and establish the best possible provision of healthcare and interventions with hundreds of daily publications is many times overwhelming. Not only that, but to guarantee preventive measures to patients and healthcare providers is key in facing Covid-19 and working in the complete chain of care, through primary care to the referral center [16, 17]. Under-resourced settings are at increased risk and challenge. Outcomes in such settings seem not as reported and reliable data is still not available to conclude.

Preparedness is based on pre-existing knowledge and resilience, although during a pandemic of a previously unknown disease, not only on its course but also on its mode and period of transmission, it is not too straightforward. Previous knowledge of similar diseases is the cornerstone helped build the response. The goals are always to prevent the intra-institutional spread and to ascertain that safe, timely and adequate care is offered to patients [18].

The pandemic is overstretching health systems worldwide, more so where the health systems have underlying fragilities. A recent report using data from the Brazilian national surveillance system on severe respiratory disease, presented a very high number of maternal deaths, with special attention to postpartum mortality and with information on delayed healthcare, since a significant number of cases did not receive respiratory support or admission to intensive care units [19], while other studies presented overall low mortality during gestation [20].

Due to the low number of performed tests among the overall population, it is possible, however, that the mortality index reported by the Brazilian study may be overestimated. Delays have already been highlighted among pregnancies for severe maternal morbidity long ago, showing a clear and significant association among frequency of delay and severity of outcome, indicating that timely and adequate management is correlated to survival [21], and the urge of a pandemic can only exacerbate such limitations.

The three delays model, by Thaddeus and Maine, defines three major components or phases: phase I–delay to seek care by the individual and/or family, phase II–delay in reaching an adequate health care facility, and phase III–delay in receiving adequate care at the health facility [22]. All these are key towards improving outcomes in COVID-19. Possibly the finding that not all participating centers had the strategy of training their health professionals on the assistance of severe acute respiratory syndrome and proceeding intubation and cardiopulmonary resuscitation could, at least in part, explain the severity level and the high associated mortality as described for the country. In the considered centers, there were 15 maternal deaths due to COVID-19, however, 12 deaths occurred in the North and Northeast regions (worst HDI- Human Development Index), what corroborates to previous data on worse outcomes in under-resourced settings.

A major limitation in many settings, as shown in the current results, is the restricted availability of testing for SARS-CoV-2 and when available, the delayed turnover of results and heterogeneity of the characteristics of the tests across centers [23–25]. This has a special impact in pregnancy, because of the needed follow-up of maternal and fetal assessment depending on gestational age, the decision on the timing of delivery and all the implications of needed protective equipment for a safe childbirth procedure, including the neonate and the corresponding health team. However, adequate or even intensive care to women with severe signs should always be timely performed even before the results of testing, avoiding phase III delays, as previously discussed. In addition, the heterogeneity of tests makes the comparison of the incidence of COVID-19 cases more difficult, which increase challenges on the response to the pandemic using such epidemiological information.

Not only that, but the impact considering suspected cases, in restriction of companionship during childbirth, delayed mother-child contact after delivery and excessive interventions, including increased risk for cesarean section. Even with growing evidence supporting the maintenance of health care that preserves a positive experience during childbirth [16], this is not always respected in many maternities worldwide.

The EAC within each facility were established to develop and manage strategies of preparedness and response, and to develop continuous planning, which includes updating policies and protocols. Moreover, the COVID-19 pandemic required changes in the maternity workforce and equipment resources. The role of the EACs and support by the REBRACO study group, in the Brazilian context, was considered important because there has been conflicting information from medical societies, local and federal governments, public health agencies, and the Brazilian Ministry of Health. Also, there have been limited resources and support available for the maternal and perinatal health area from national agencies. Up to August/2020, there was no national guidance specific for pregnancy. Coordinated efforts should be the rule for adequate support to health facilities [26].

Previous experience, during the Influenza A H1N1 pdm09 epidemic showed that infection during pregnancy was associated with severe maternal morbidity (SMM) and, consequently, with increased adverse perinatal outcomes such as low birth weight, preterm deliveries, still-births, neonatal and maternal mortality [27]. The availability of ICU and proper management of maternal morbidity cases were associated with a reduction of maternal near-miss and maternal deaths among women with SMM [28]. Nevertheless, there is current evidence that a significant proportion of pregnant women who died due to COVID-19 in Brazil did not have access to an ICU [19]. Providing adequate training, assuring safe workplaces and sufficient intensive care units remain unmet needs in the fight against the pandemic in Brazil.

The qualitative approach has raised interesting experiences related to the development and implementation of an emergency action committee to respond to the pandemic in obstetric units. The health managers described how fear has affected all stages of the response to the

pandemic, including preparedness and implementation of protocols and the daily attendance routine in obstetric units. The "pandemic fear" phenomenon has been reported worldwide in the context of the COVID-19 and the healthcare services [29–31]. The absence of effective treatment, non-availability of vaccine during the first wave, and the possibility of a collapse in the health system has only worsen the scenario [32]. The response of health care services towards the pandemic affects how people experience and react to challenges [29, 33].

In fact, some key components for rapid hospital readiness involve leadership, coordination, communication, and community engagement [26]. Without these components, people, including health professionals, may feel unsupported. The provision of these skills may enhance the ability of health facilities in organizing their crisis management plans and to ensure informed risk analyses, decision-making strategies, and confidence amongst all hospital staff and stakeholders. Also, a multi-strategy approach is essential to provide mental health support to health professionals during the pandemic, including multidisciplinary mental health teams, clear communication involving regular and accurate updates on the COVID-19 outbreak and safe psychological counselling services (e.g. via electronic devices or apps) [32, 34]. The institutions considered in this study, mostly university hospitals and referral centers, had availability for such support, what might not be true or representative of most maternities in Brazil.

The healthcare managers are in a leadership position, which demands dealing with challenging dilemmas such as lack of health professionals, fear of the staff in working in the front-line and lack of resources at the very same moment when the workforce and such resources are needed the most. Dealing with extra work and with personal issues and worries requires resiliency, persistency and the ability to adapt and listening. Improving health care leadership is of great importance as part of the response to the pandemic, especially when there are conflicting policies by governors and policy-makers [3].

As a limitation of the current REBRACO study, data from the 16 maternity centers included is not representative of the real maternal health care throughout Brazil, a country of continental size and major disparities. Therefore, the presented results do not provide data to ascertain improved clinical outcomes among considered maternities. In addition, as previously informed, the centers did not follow a common standard protocol for clinically dealing with cases of Covid19 during pregnancy, because only very recently the Ministry of Health issued general recommendations for such management.

Details on how maternities prepared to face the Covid-19, with quantitative and qualitative approaches in a middle-income setting that was severely affected by the disease, using institutions that are part of a National Network for Studies in Reproductive and Perinatal Health can help health managers and professionals in facing not only the current but also future challenges. Additionally, prospective evaluation on short and long-term clinical and social determinants of the coronavirus disease in Brazil will add to epidemiological reports [35, 36], the main dataset available so far in the country.

## Conclusions

Study findings suggest that maternities of the REBRACO initiative underwent major changes in facing the pandemic, with limitations on testing, difficulties in infrastructure and human resources. Leadership, continuous training, implementation of evidence-based protocols and collaborative initiatives are key to transpose the fear of the virus and ascertain adequate healthcare inside maternities, especially in low and middle-income settings. Policy makers need to address the specificities in considering reproductive health and childbirth during the COVID-19 pandemic and prioritize research and timely testing availability.

## Supporting information

**S1 Checklist. STROBE statement—checklist of items that should be included in reports of observational studies.**
(DOCX)

**S1 Data. Variables requested for the characterization of the included REBRACO centers.**
List of variables which were collected through online forms.
(DOCX)

**S2 Data. Data collection form.** Information about the organization, health care and consolidated results of the REBRACO participating centers (from March to August).
(DOCX)

**S3 Data. Interview script.** Semi-structured interview script for health managers, used on the qualitative study.
(DOCX)

## Acknowledgments

We would like to recognize the participation of the other members from the REBRACO Study Group leaded by Jose G Cecatti (cecatti@unicamp.br): Sherly Metelus[1], Amanda D Silva[1], Paulo S R Junior[1], Thais G Sardinha[1], Rodolfo R Japenga[1], Erica R F Urquiza[1], Maíra R Machado[1], Marcela Maria Simões[1], Larissa M Solda[1], Patrícia B Peres[2], Cristiane L Arbeli[2], Rafael M Quevedo[2], Carolina F Yamashita[2], Julia D Corradin[2], Isabella Bergamini[2], José Geraldo L Ramos[3], Maria Lúcia R Oppermann[3], Laisa S Quadro[3], Lina Marins[3], Érika V Paniz[3], Aline C Costa[4], Marina HL Almeida[4], Bruna FV Moura[4], Lidiane R França[4], Hanna Vieira[4], Rafael B Aquino[4], Daisy Lucena[5], Feitosa L Pinheiro[5], Denise H F Cordeiro[5], Priscila L Miná[5], Carol Dornellas[5], Evelyn Traina[6], Sue Yazaki-Sun[6], Priscilla Mota[6], Arimaza C Soares[6], Ellen Machado[7], Anne Bergmann[7], Gustavo Raupp Santos[7], Aline Tosetto[8], Sabrina Savazoni[8], Bruna E Parreira[9], Rayra AM Maciel[11], Caio RV Leal[11], Marcos Nakamura-Pereira[12], Bruna O Guerra[12], Gabriela Gorga[12], Kevin FA Oliveira[13], Débora F Leite[14], Isabella Monteiro[14], Cristiane O Santos[15], Marina M dos Santos[15], Carlos Neto[15], Thiago Gomes[15], Isabela R Pereira[16], Clélia A Salustrino[16], Valéria B Pontes[16], Roberto AS Franco[16], João P Bilibio[16], Gislânia PF Brito[16], Hana PC Pinto[16], Danielle L Oliveira[16], Andrezza A Guerra[16], Andrea O Moura[16], Natasha Pantoja[16], Fernanda David[16,] Alina Silva[16]. Also, we would like to acknowledge the staff from the coordinator centre which has had a major contribution as part of the REBRACO initiative: Angela M Bacha, Anderson Borovac-Pinheiro, Belmiro G Pereiro, Fernanda G Surita, Eliana M Amaral, Elton C Ferreira, Helaine M Milanez, Jamil P S Caldas, Luis Bahamondes, Luiz F Baccaro, Marcelo Nomura, Patrícia M Rehder, Renata Zacarias Simone, Renato Passini Jr, Sergio T Marba and Tábata R Zumpano Santos.

## Author Contributions

**Conceptualization:** Maria Laura Costa, Renato T. Souza, Rodolfo C. Pacagnella, Silvana F. Bento, Samira M. Haddad, Jose G. Cecatti.

**Data curation:** Maria Laura Costa, Renato T. Souza, Carolina C. Ribeiro-do-Valle, Giuliane J. Lajos, Guilherme M. Nobrega, Thayna B. Griggio, Charles M. Charles, Ricardo P. Tedesco, Karayna G. Fernandes, Sérgio H. A. Martins-Costa, Frederico J. A. Peret, Francisco E. Feitosa, Rosiane Mattar, Edson V. Cunha Filho, Janete Vetorazzi, Samira M. Haddad, Carla B.

Andreucci, José P. Guida, Mário D. Correa Junior, Marcos A. B. Dias, Leandro G. Oliveira, Elias F. Melo Junior, Carlos A. S. Menezes, Marília G. Q. Luz.

**Formal analysis:** Maria Laura Costa, Renato T. Souza, Silvana F. Bento, Guilherme M. Nobrega, Thayna B. Griggio.

**Funding acquisition:** Maria Laura Costa.

**Investigation:** Renato T. Souza, Rodolfo C. Pacagnella, Silvana F. Bento, Carolina C. Ribeiro-do-Valle, Adriana G. Luz, Giuliane J. Lajos, Guilherme M. Nobrega, Thayna B. Griggio, Charles M. Charles, Ricardo P. Tedesco, Karayna G. Fernandes, Sérgio H. A. Martins-Costa, Frederico J. A. Peret, Francisco E. Feitosa, Rosiane Mattar, Edson V. Cunha Filho, Janete Vetorazzi, Carla B. Andreucci, José P. Guida, Mário D. Correa Junior, Marcos A. B. Dias, Leandro G. Oliveira, Elias F. Melo Junior, Carlos A. S. Menezes, Marília G. Q. Luz, Jose G. Cecatti.

**Methodology:** Maria Laura Costa, Renato T. Souza, Rodolfo C. Pacagnella, Silvana F. Bento, Adriana G. Luz, Giuliane J. Lajos, Guilherme M. Nobrega, Thayna B. Griggio, Charles M. Charles, Karayna G. Fernandes, José P. Guida, Jose G. Cecatti.

**Project administration:** Maria Laura Costa, Renato T. Souza, Rodolfo C. Pacagnella, Silvana F. Bento, Jose G. Cecatti.

**Resources:** Maria Laura Costa.

**Software:** Silvana F. Bento, Guilherme M. Nobrega.

**Supervision:** Renato T. Souza, Jose G. Cecatti.

**Validation:** Guilherme M. Nobrega, Samira M. Haddad, Jose G. Cecatti.

**Writing – original draft:** Maria Laura Costa, Renato T. Souza, Silvana F. Bento.

**Writing – review & editing:** Rodolfo C. Pacagnella, Carolina C. Ribeiro-do-Valle, Adriana G. Luz, Giuliane J. Lajos, Guilherme M. Nobrega, Thayna B. Griggio, Charles M. Charles, Ricardo P. Tedesco, Karayna G. Fernandes, Sérgio H. A. Martins-Costa, Frederico J. A. Peret, Francisco E. Feitosa, Rosiane Mattar, Edson V. Cunha Filho, Janete Vetorazzi, Samira M. Haddad, Carla B. Andreucci, José P. Guida, Mário D. Correa Junior, Marcos A. B. Dias, Leandro G. Oliveira, Elias F. Melo Junior, Carlos A. S. Menezes, Marília G. Q. Luz, Jose G. Cecatti.

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
