## [Decision Letter · Decision Letter 0]

5 Jan 2021

PONE-D-20-29941

Facing the COVID-19 Pandemic inside maternities in Brazil: how to overcome challenges towards adequate careFacing the COVID-19 Pandemic inside maternities in Brazil: how to overcome challenges towards adequate care

PLOS ONE

Dear Dr. Costa,

Thank you for submitting your manuscript to PLOS ONE. After careful consideration, we feel that it has merit but does not fully meet PLOS ONE’s publication criteria as it currently stands. Therefore, we invite you to submit a revised version of the manuscript that addresses the points raised during the review process.

Please read carefully all comments below before resubmitting your paper. 

We look forward to receiving your revised manuscript.

Kind regards,

Marzia Lazzerini, PhD

Academic Editor

PLOS ONE

Journal Requirements:

2. Thank you for including your ethics statement: "Ethical approval 81 for the main study and in each participating center was obtained (Letter of Approval numbers 82 4.047.168, 4.179.679, and 4.083.988)."   

 a.Please amend your current ethics statement to include the full name of the ethics committee/institutional review board(s) that approved your specific study.

 b.Once you have amended this/these statement(s) in the Methods section of the manuscript, please add the same text to the “Ethics Statement” field of the submission form (via “Edit Submission”).

3. For your cross-sectional study, please provide additional details regarding participant consent. In the ethics statement in the Methods and online submission information, please ensure that you have specified (1) whether consent was obtained for the questionnaire (2) whether consent was informed and (2) what type you obtained (for instance, written or verbal, and if verbal, how it was documented and witnessed).  If the need for consent was waived, please ensure that you have discussed whether all data were fully anonymized before you accessed them and/or whether the IRB or ethics committee waived the requirement for informed consent.

4. Regarding verbal consent in your qualitative study, in the Methods, please state:

- Why written consent could not be obtained

- Whether the Institutional Review Board (IRB) approved use of oral consent

- How oral consent was documented

For more information, please see our guidelines for human subjects research: https://journals.plos.org/plosone/s/submission-guidelines#loc-human-subjects-research

5. In your Methods section, please provide additional information about the participant recruitment method and the demographic details of your participants. Please ensure you have provided sufficient details to replicate the analyses such as: a) the recruitment date range (month and year), b) a description of any inclusion/exclusion criteria that were applied to participant recruitment, c) a table of relevant demographic details, d) a statement as to whether your sample can be considered representative of a larger population.

6. Please include a copy of the interview guide used in the qualitative study, in both the original language and English, as Supporting Information, or include a citation if it has been published previously.

7. Please include additional information regarding the survey or questionnaire used in the cross-sectional study and ensure that you have provided sufficient details that others could replicate the analyses. For instance, if you developed a questionnaire as part of this study and it is not under a copyright more restrictive than CC-BY, please include a copy, in both the original language and English, as Supporting Information.

8. Please include a discussion of any limitations to your study in the Discussion section.

9. To comply with PLOS ONE submission guidelines, in your Methods section, please provide additional information regarding your statistical analyses. For more information on PLOS ONE's expectations for statistical reporting, please see https://journals.plos.org/plosone/s/submission-guidelines.#loc-statistical-reporting.

10.Thank you for stating the following in the Acknowledgments Section of your manuscript:

"The study was supported by FAEPEX-Unicamp (Fundo de Apoio ao Ensino, à Pesquisa e à Extensão) under the grant number 2300/20."

11. Please amend either the title on the online submission form (via Edit Submission) or the title in the manuscript so that they are identical.

12. One of the noted authors is a group or consortium REBRACO Study Group#. In addition to naming the author group, please list the individual authors and affiliations within this group in the acknowledgments section of your manuscript. Please also indicate clearly a lead author for this group along with a contact email address.

13.We note that [Figure(s) 1] in your submission contain map images which may be copyrighted. All PLOS content is published under the Creative Commons Attribution License (CC BY 4.0), which means that the manuscript, images, and Supporting Information files will be freely available online, and any third party is permitted to access, download, copy, distribute, and use these materials in any way, even commercially, with proper attribution. For these reasons, we cannot publish previously copyrighted maps or satellite images created using proprietary data, such as Google software (Google Maps, Street View, and Earth). For more information, see our copyright guidelines: http://journals.plos.org/plosone/s/licenses-and-copyright.

a.    You may seek permission from the original copyright holder of Figure(s) [1] to publish the content specifically under the CC BY 4.0 license. 

Additional Editor Comments (if provided):

Please address carefully the points below:

1) Clarify (abstract and method section) the study design: is this an observational mixed method study? was there an intervention component?

2) Move the description of the REBRACO initiative (lines 381- 404) in the method section, clarifying to which extent there was an intervention, and when. Do separate activities which related to the study period from activities which has been undertaken after this study

3) Refer to the appropriate guidelines of reporting based on the study design (see https://www.equator-network.org/reporting-guidelines/)

4) Consider revising the title based on the study design and findings, ie, this specific manuscript does not seem to directly aim at exploring how to overcome challenges to adequate care (This would require an intervention study), neither it shows that challenges has been overcomed (table 2) - rather it seems to provide an analysis of the challenges faced by hospitals in a research network and related indicators

5) Clarify the case definition for "MD due to COVID", and, if possible, add details on case characteristics. Adequately comment in the discussion section: how this MD rate compare to the usual MD rates in the included hospitals? where cases clustered or equally distributed among hospitals? where MD most probably due to COVID our to low quality of care?

6) Ensure that all statements in the discussion section have an appropriate reference (eg, line 327 the reference on the Brazilian epidemiological reports is lacking; line 353 the reference open the recent report from Brazil is also missing; line 405 the statement on "major limitation in many settings" miss a reference; please do revise the whole discussion section)

7) Improve the quality of English (eg ask a native English speaker to revise the manuscript)

8) Revise the manuscript according Plos guidelines https://journals.plos.org/plosone/s/submission-guidelines

Reviewers' comments:

Reviewer's Responses to Questions

**Comments to the Author**

1. Is the manuscript technically sound, and do the data support the conclusions?

Reviewer #1: Yes

Reviewer #2: Partly

2. Has the statistical analysis been performed appropriately and rigorously? 

Reviewer #1: I Don't Know

Reviewer #2: N/A

3. Have the authors made all data underlying the findings in their manuscript fully available?

Reviewer #1: Yes

Reviewer #2: Yes

4. Is the manuscript presented in an intelligible fashion and written in standard English?

Reviewer #1: Yes

Reviewer #2: Yes

5. Review Comments to the Author

Reviewer #1: Good relevant study with such substantial information added on how to deal with pandemic especially in Low-middle and relevant to high income economies.

Few queries:

Testing challenges; It is mentioned testing was performed full stop; but no further information on how testing was done. For lessons to other parts of the world, it would best to describe how testing was performed. Done by patients themselves or health providers? Inline with testing, any particular lessons we can learn to why there were delays in getting back the results in some centres? It can be questionable if attributable to low resources or fear of sampling and analysis.

Number of cases, challenges and strategies: 29 death attributable to COVD-19; any particular risk factors, co-morbidities ?

Reviewer #2: While the work of the authors in preparing their centers for the care of pregnant women is laudable and to be commended, I am not sure how this paper, other than by describing the REBRACO program, provides any evidence that such a program provided for improved care and led to reduced maternal/neonatal morbidity or mortality. There are no comparisons to any non-REBRACO participating centers and there isn't even a comparison of the maternal death rate during the pandemic months in the REBRACO centres as compared to, say, the same centers during a similar time period before the pandemic. Many such multicenter programs have been set up world wide to better share scarce resources (i.e., PPE), coordinate referrals, better communicate clinical advances and risks and to best document clinical outcomes. While the authors have well reported the clinical outcomes, there needs to be some determination as to whether this approach actually provided for improved clinical outcomes.

6. PLOS authors have the option to publish the peer review history of their article (what does this mean?). If published, this will include your full peer review and any attached files.

Reviewer #1: No

Reviewer #2: No

---

## [Author Response · Author response to Decision Letter 0]

22 Jan 2021

Dear editor,

We acknowledge the detailed evaluation of our study; points raised by the reviewers were important and helped to improve our manuscript.

Bellow, you will find a point-by-point response answering reviewer’s comments. All changes are highlighted (in yellow) in the manuscript.

Sincerely,

Maria Laura Costa, MD, PhD

 RESPONSE: Thank you. We have checked the style requirements and made the needed modifications. 

2. Thank you for including your ethics statement: "Ethical approval for the main study and in each participating center was obtained (Letter of Approval numbers 4.047.168, 4.179.679, and 4.083.988)." 

 a.Please amend your current ethics statement to include the full name of the ethics committee/institutional review board(s) that approved your specific study.

 b.Once you have amended this/these statement(s) in the Methods section of the manuscript, please add the same text to the “Ethics Statement” field of the submission form (via “Edit Submission”).

RESPONSE: The full name of the ethics committee/institutional review boards that approved the study were included- page 5:

“Ethical approval for the main study and in each participating center was obtained (Letter of Approval numbers 4.047.168, 4.179.679, and 4.083.988, from the coordinating center´s Institutional Review Board from the University of Campinas, the IRB of the School of Medicine of Jundiai, IRB of the Clinics Hospital of Porto Alegre, the IRB of the Unimed Maternity of Belo Horizonte, the IRB of MEAC from the Federal University of Ceara, the IRB of the Federal University of Sao Paulo, IRB of Moinhos de Vento Hospital from Porto Alegre, IRB of the Jorge Rossmann Regional Hospital from Itanhaem, IRB of the Federal University of Sao Carlos, IRB of the State Hospital of Sumare, IR of the Feral University of Minas Gerais, IRB of the Fernandes Figueira Institute from Fiocruz in Rio de Janeiro, IRB of the School of Medicine from the State University of Sao Paulo in Botucatu, IRB of the Federal University of Pernambuco, IRB of the Climerio de Oliveira Maternity from the Federal University of Bahia in Salvador, and the IRB of the Santa Casa de Misericordia do Pará from Belem.”

 3. For your cross-sectional study, please provide additional details regarding participant consent. In the ethics statement in the Methods and online submission information, please ensure that you have specified (1) whether consent was obtained for the questionnaire (2) whether consent was informed and (2) what type you obtained (for instance, written or verbal, and if verbal, how it was documented and witnessed). If the need for consent was waived, please ensure that you have discussed whether all data were fully anonymized before you accessed them and/or whether the IRB or ethics committee waived the requirement for informed consent.

4. Regarding verbal consent in your qualitative study, in the Methods, please state:

- Why written consent could not be obtained

- Whether the Institutional Review Board (IRB) approved use of oral consent

- How oral consent was documented

For more information, please see our guidelines for human subjects research: https://journals.plos.org/plosone/s/submission-guidelines#loc-human-subjects-research

 RESPONSE: Considering questions 3 and 4- Thank you for your comment and questions. This manuscript represents a great study initiative in Brazil, called REBRACO, with many aims to broadly understand the impact of the COVID-19 in our scenario. Therefore, we have split this major project in three, for ethical approval, dividing the cross-sectional, cohort of suspected cases and qualitative approaches. That is why we presented 3 ethical approval numbers above. We have authorization for both- written informed consent and consent by phone, with authorized recording. Mostly suspected cases with hospital admissions have written informed consent, cases that were evaluated and tested in the emergency room, with no needed admission were later consented by phone. Qualitative studies, especially the one presented here, with health professionals, were done remotely, by phone, since maternities considered are in different parts of the country and all interviews were performed by the same researcher in the coordinating center. 

We would like to acknowledge the fact that the current study was based on aggregated data collected via the data collection form that was made available as supporting information and that individual data related to the number of suspected and/or confirmed COVID-19 cases and the related clinical characterization, rate of progression to acute respiratory distress syndrome, and mortality will be properly and timely addressed in another publication of the Consortium.

The information is now clearer in the manuscript, methods, pages 6-7:

“All suspected cases of COVID-19 in the considered maternities were assessed for an informed consent and their data was only included after such consent. The data sharing information on the questionnaire of readiness towards the pandemic and overall numbers of each institution were approved by the coordinating center and locally in each participating center. Data was stored in specific database created for this study and protected by password in a de-identified way. For mild cases not admitted to the hospital, considering biosafety reasons, informed consent by telephone contact was approved by the Institutional Review Board. Participant identity and all personal data as well will remain confidential.”

“We also performed a qualitative component, considering key obstetricians involved in the decision making of each institution, to listen to their perspectives on the challenges to face the pandemic in their setting. Qualitative study by purposeful sampling and saturation sampling was undertaken after individual verbal consent. All semi-structured interviews were performed by a skilled social researcher with experience in the field and were recorded on audio. For this approach, the Institutional Review Board (IRB) approved use of oral consent. Maternities considered are in different regions of the country and all interviews were performed by the same researcher in the coordinating center, by telephone, with a specific software for further transcription and analysis (“ReShape”), protected by password in a de-identified manner.”

5. In your Methods section, please provide additional information about the participant recruitment method and the demographic details of your participants. Please ensure you have provided sufficient details to replicate the analyses such as: a) the recruitment date range (month and year), b) a description of any inclusion/exclusion criteria that were applied to participant recruitment, c) a table of relevant demographic details, d) a statement as to whether your sample can be considered representative of a larger population.

RESPONSE: thank you for your comment, we understand that it was not clear that we here present part of a major study called REBRACO. We have included a paragraph to ascertain that this is now accurate- page 3

“The REBRACO study aims to broadly understand the impact of the COVID-19 in the considered institutions, evaluating clinical, epidemiological and laboratory aspects of SARS-CoV-2 infection during pregnancy and the postpartum (11). The current analysis presents data on how centers faced the pandemic and prepared each institution for the identification and medical care of patients, considering information from January to July 31, 2020.” 

Considering the selection of cases for the qualitative component of the current analysis, the local PI identified key health professionals/obstetricians involved in the decision making of each institution and/or responsible for the maternities’ response towards the pandemic. The phone contact to obtain ethical informed consent and the interview were performed by the same research assistant, from the coordinating center. We have added a sentence to the paragraph in page 6:

“We also performed a qualitative component, considering key obstetricians involved in the decision making of each institution, to listen to their perspectives on the challenges to face the pandemic in their setting. These participants were identified by local PIs (principal investigators).” 

The details on the cohort of suspected/positive COVID-19 cases, with demographic, clinical, maternal and perinatal outcomes will be further presented in a future analysis. We have included in the discussion, relevant information about Brazil, with data from published studies (from National surveillance system) analyzing clinical characteristics and risk factors for COVID-19-related maternal deaths in the country, that showed the postpartum period, obesity, diabetes, cardiovascular disease and black ethnicity as conditions associated with the fatality rate. 

The REBRACO initiative includes 16 maternity hospitals, in 4 of the five regions of the country, mostly University Hospitals, with intensive care support and referral centers for maternal healthcare. Therefore, we understand that it is not representative of the country. Brazil is a country of continental size and most of our centers concentrate in the southeast region. We have included a paragraph in the discussion to add this information. 

“The institutions considered in this study, mostly university hospitals and referral centers, had availability for such support, what might not be true or representative of most maternities in Brazil.”

6. Please include a copy of the interview guide used in the qualitative study, in both the original language and English, as Supporting Information, or include a citation if it has been published previously.

RESPONSE: Thank you for the suggestion. We included the semi-structured interview script in both languages (Portuguese and English) as supporting information (S2). The respective reference to the document was included in the methods (Page 3).

7. Please include additional information regarding the survey or questionnaire used in the cross-sectional study and ensure that you have provided sufficient details that others could replicate the analyses. For instance, if you developed a questionnaire as part of this study and it is not under a copyright more restrictive than CC-BY, please include a copy, in both the original language and English, as Supporting Information.

RESPONSE: Thank you for your suggestion. In addition to the detailed description of variables included in the data collection form, we included the data collection form that was sent to the health managers as Supporting Information. This information was included in the methods (Page 5-6).

“The data collection form is provided as Supporting Information (S1). Data were collected with the assistance of the maternity hospital staff. We conducted a descriptive analysis of the information above, including the characteristics of the maternities, their response program, the training program and some indicators related to the COVID-19 pandemic such as the number of cases, number of maternal deaths, maternal mortality ratio due to COVID-19 in the period, time interval between the implementation of the response program and the first suspected COVID-19 cases in each center and the characteristics of the tests available in the obstetric units.”

8. Please include a discussion of any limitations to your study in the Discussion section.

RESPONSE: We have included a paragraph in the discussion, with the main identified limitations of the study. Page 24:

“As a limitation of the current REBRACO study, data from the 16 maternity centers included is not representative of the real maternal health care throughout Brazil, a country of continental size and major disparities. Therefore, the presented results do not provide data to ascertain improved clinical outcomes among considered maternities. In addition, as previously informed, the centers did not follow a common standard protocol for clinically dealing with cases of Covid19 during pregnancy, because only very recently the Ministry of Health issued general recommendations for such management.”

. To comply with PLOS ONE submission guidelines, in your Methods section, please provide additional information regarding your statistical analyses. For more information on PLOS ONE's expectations for statistical reporting, please see https://journals.plos.org/plosone/s/submission-guidelines.#loc-statistical-reporting.

RESPONSE: Thank you for the suggestion. The following information was added to the manuscript in the methods (Page 5-6):

“We conducted a descriptive analysis of the information above, including the characteristics of the maternities, their response program, the training program and some indicators related to the COVID-19 pandemic such as the number of cases, number of maternal deaths, maternal mortality ratio due to COVID-19 in the period, time interval between the implementation of the response program and the first suspected COVID-19 cases in each center and the characteristics of the tests available in the obstetric units.”

“Thematic analysis was finally carried out [12] after the familiarization with the collected data. Then we identified the main themes after generating the initial codes. The themes were reviewed and, finally, they were defined and named according to the main challenges identified in the interviewees’ narratives, with their corresponding experiences, conflicts and answers/actions.”

We would like to acknowledge the fact that we only conducted descriptive and thematic analyses for the quantitative and qualitative components of the study, respectively. 

10.Thank you for stating the following in the Acknowledgments Section of your manuscript:

"The study was supported by FAEPEX-Unicamp (Fundo de Apoio ao Ensino, à Pesquisa e à Extensão) under the grant number 2300/20."

 RESPONSE: Thank you. We have now corrected the above text and removed funding-related information from acknowledgements and included the amended statements within our cover letter.

11. Please amend either the title on the online submission form (via Edit Submission) or the title in the manuscript so that they are identical.

 RESPONSE: We are sorry. The title was corrected in the submission.

12. One of the noted authors is a group or consortium REBRACO Study Group#. In addition to naming the author group, please list the individual authors and affiliations within this group in the acknowledgments section of your manuscript. Please also indicate clearly a lead author for this group along with a contact email address.

RESPONSE: The lead author for this group is Jose Guilherme Cecatti, the last author of this manuscript. 

The individual authors and affiliation and the indication of the lead author for each group with a contact e-email address within the respective group were added in the acknowledgments accordingly as suggested.

 13.We note that [Figure(s) 1] in your submission contain map images which may be copyrighted. All PLOS content is published under the Creative Commons Attribution License (CC BY 4.0), which means that the manuscript, images, and Supporting Information files will be freely available online, and any third party is permitted to access, download, copy, distribute, and use these materials in any way, even commercially, with proper attribution. For these reasons, we cannot publish previously copyrighted maps or satellite images created using proprietary data, such as Google software (Google Maps, Street View, and Earth). For more information, see our copyright guidelines: http://journals.plos.org/plosone/s/licenses-and-copyright.

 a. You may seek permission from the original copyright holder of Figure(s) [1] to publish the content specifically under the CC BY 4.0 license. 

RESPONSE: Figure 1 was adapted from a map contained in the Brazilian public domain database – Portal Domínio Público (http://www.dominiopublico.gov.br). Thus, the image has no copyright linked to be evidenced. We have included this information in the Figure footnote

Additional Editor Comments (if provided):

Please address carefully the points below:

1) Clarify (abstract and method section) the study design: is this an observational mixed method study? was there an intervention component?

RESPONSE: Thank you. This is an observational mixed method study. We have made that clear in the abstract and methods. There were no interventions.

2) Move the description of the REBRACO initiative (lines 381- 404) in the method section, clarifying to which extent there was an intervention, and when. Do separate activities which related to the study period from activities which has been undertaken after this study

RESPONSE: Thank you. This paragraph is important to add information on the activities of the REBRACO study group. The content of these meetings have inspired the current manuscript, as we learned with each of the local center´s presenting their challenges and ways to overcome them. We have summarized the paragraph and made it clear that there were no interventions. It is now the last paragraph of the methods. 

“In the REBRACO initiative, there are no planned interventions in each included centers. All the research team, including local-PIs, health professionals involved in the local Emergency Action Committees, and other collaborators (Ob&Gyn residents, consultants, etc.) from all the included centers, have been invited to participate in weekly virtual meetings, sharing information regarding the organization of the different health services, the barriers and facilitators in the implementation of healthcare and training of health professionals in each center and discussions on current findings on maternal and perinatal outcomes during pregnancy and postpartum of COVID-19 infected women. These meetings have inspired the current analysis.” 

3) Refer to the appropriate guidelines of reporting based on the study design (see https://www.equator-network.org/reporting-guidelines/)

RESPONSE: STROBE guideline was followed and included as a supplementary material. 

4) Consider revising the title based on the study design and findings, ie, this specific manuscript does not seem to directly aim at exploring how to overcome challenges to adequate care (This would require an intervention study), neither it shows that challenges has been overcomed (table 2) - rather it seems to provide an analysis of the challenges faced by hospitals in a research network and related indicators

RESPONSE: Thank you. You are correct, we have changed the title according to the above suggestion. It now reads: “Facing the COVID-19 Pandemic inside maternities in Brazil: challenges towards adequate care in the REBRACO initiative”

5) Clarify the case definition for "MD due to COVID", and, if possible, add details on case characteristics. Adequately comment in the discussion section: how this MD rate compare to the usual MD rates in the included hospitals? where cases clustered or equally distributed among hospitals? where MD most probably due to COVID our to low quality of care?

RESPONSE: Thank you for the comment and questions. The focus of the current analysis is based on the topics presented. We intend to further detail maternal deaths due to COVID in another analysis and that is why we did not present data specific to reported cases, such as sociodemographic information, clinical background, maternal and perinatal outcomes. We here presented overall numbers, to provide a better picture of the impact of the pandemic. There were 15 maternal deaths due to COVID-19 in the participating centers during the study period. The 29 maternal deaths were due to all causes (and the main cause in Brazil is hypertension). This is described in page 7. Each participating center presented its data according to the information in medical charts.

“There were overall 29 maternal deaths (distributed in only 7 of the 16 centers) and 15 (51.8%) attributed to COVID-19 (in 6 centers), with a Maternal mortality rate (MMR) of 127.8/100.000 LB.” 

The discussion if MD were most probably due to COVID or to low quality of care is very complex and we cannot answer it yet, in this manuscript. However, it is important to consider published information about Brazil, with data from our national surveillance system. Data on maternal mortality due to COVID-19 in Brazil showed that up to July 14, 2020, 2,475 cases of COVID‐19 acute respiratory distress syndrome (ARDS) were notified, and among them 204 maternal deaths occurred. Delays in receiving care may have played an important role in fatality rate due to COVID19 among pregnant/postpartum women in our country. Initial “stay at home” recommendation and restrictions on public transportation lengthened hospital admission, characterizing the first delay. Social vulnerability may have led to delays on reaching ICUs, corresponding to the second delay. Lack of thromboprophylaxis after cesarean in high‐risk pregnant women may be an example of the third delay (Menezes, M.O., Takemoto, M.L.S., Nakamura‐Pereira, M., Katz, L., Amorim, M.M.R., Salgado, H.O., Melo, A., Diniz, C.S.G., de Sousa, L.A.R., Magalhaes, C.G., Knobel, R., Andreucci, C.B. and (2020), Risk factors for adverse outcomes among pregnant and postpartum women with acute respiratory distress syndrome due to COVID‐19 in Brazil. Int. J. Gynecol. Obstet., 151: 415-423). We believe that maternal deaths are related both to the severity of coronavirus disease upon obstetric patients, but also due to inappropriate care, including delays. 

However, comparing with nationwide data, we understand that the REBRACO study group does not represent healthcare in the country, with most centers in the southeast region and with better maternal and perinatal outcomes, since most centers are University Hospitals, with availability of Intensive Care Support. However, we have highlighted in the discussion, that most cases of MD (12 deaths)- occurred in the North and Northeast regions (worst HDI- Human Development Index), what corroborates to previous data on worse outcomes in under-resourced settings (page 22)

6) Ensure that all statements in the discussion section have an appropriate reference (eg, line 327 the reference on the Brazilian epidemiological reports is lacking; line 353 the reference open the recent report from Brazil is also missing; line 405 the statement on "major limitation in many settings" miss a reference; please do revise the whole discussion section)

Response: The respective references were added and the final list of references was updated. We also revised the discussion section accordingly.

7) Improve the quality of English (eg ask a native English speaker to revise the manuscript)

Response: English has been revised. 

8) Revise the manuscript according Plos guidelines https://journals.plos.org/plosone/s/submission-guidelines

Reviewers' comments:

Reviewer's Responses to Questions

Comments to the Author

1. Is the manuscript technically sound, and do the data support the conclusions?

Reviewer #1: Yes

Reviewer #2: Partly

2. Has the statistical analysis been performed appropriately and rigorously?

Reviewer #1: I Don't Know

Reviewer #2: N/A

3. Have the authors made all data underlying the findings in their manuscript fully available?

Reviewer #1: Yes

Reviewer #2: Yes

4. Is the manuscript presented in an intelligible fashion and written in standard English?

Reviewer #1: Yes

Reviewer #2: Yes

5. Review Comments to the Author

Reviewer #1: Good relevant study with such substantial information added on how to deal with pandemic especially in Low-middle and relevant to high income economies.

Few queries:

Testing challenges; It is mentioned testing was performed full stop; but no further information on how testing was done. For lessons to other parts of the world, it would best to describe how testing was performed. Done by patients themselves or health providers? Inline with testing, any particular lessons we can learn to why there were delays in getting back the results in some centres? It can be questionable if attributable to low resources or fear of sampling and analysis.

RESPONSE: We included further information about the tests that had been performed in the REBRACO centers during the study period. The following sentence and the respective Figure was included in the results (Page 9).

“Figure 2 shows the characteristics of the tests performed in the REBRACO centers. In half of the centers (n=8), the tests were not performed in the facility and the characteristics were not available. The characteristic of the RT-PCR and serological tests available were heterogeneous across centers.”

The last paragraph of page 22 and first paragraph of the following page addresses the topics raised by the reviewers.

“A major limitation in many settings, as shown in the current results, is the restricted availability of testing for SARS-CoV-2 and when available, the delayed turnover of results and heterogeneity of the characteristics of the tests across centers.” 

“In addition, the heterogeneity of tests make the comparison of the incidence of exposed women to COVID-19 more difficult, which increase challenges on the response to the pandemic using such epidemiological information.”

Number of cases, challenges and strategies: 29 death attributable to COVD-19; any particular risk factors, co-morbidities ?

RESPONSE: Actually, there were 15 maternal deaths due to COVID-19 in the participating centers during the study period. The 29 maternal deaths were due to all causes (and the main cause in Brazil is hypertension). This is described in page 7:

“There were overall 29 maternal deaths (distributed in only 7 of the 16 centers) and 15 (51.8%) attributed to COVID-19 (in 6 centers), with a Maternal mortality rate (MMR) of 127.8/100.000 LB.” 

We would like to clarify that we intend to explore the maternal deaths in detail in another analysis which includes the presence of risk-factors, co-morbidities, management characteristics, etc. We thank you for your understanding. However, we have improved the discussion on the scenario and numbers of maternal deaths due to COVID-19 in Brazil and delays associated to such numbers. 

Reviewer #2: While the work of the authors in preparing their centers for the care of pregnant women is laudable and to be commended, I am not sure how this paper, other than by describing the REBRACO program, provides any evidence that such a program provided for improved care and led to reduced maternal/neonatal morbidity or mortality. There are no comparisons to any non-REBRACO participating centers and there isn't even a comparison of the maternal death rate during the pandemic months in the REBRACO centres as compared to, say, the same centers during a similar time period before the pandemic. Many such multicenter programs have been set up world wide to better share scarce resources (i.e., PPE), coordinate referrals, better communicate clinical advances and risks and to best document clinical outcomes. While the authors have well reported the clinical outcomes, there needs to be some determination as to whether this approach actually provided for improved clinical outcomes.

RESPONSE: Thank you for this consideration. We understand the title was inadequate and misleading, since there are no interventions reported or analysis to compare outcomes of centers in the REBRACO and out of the research initiative. Our aim was to report challenges on facing the pandemic from centers in a middle-income setting. We presented quantitative and qualitative data to complement the analysis and this is one of the strengths of the paper. At this point, we cannot ascertain that this approach actually provided for improved clinical outcomes, however, we do believe that the participation in a committed group, including health managers, with weekly meetings on the topic and shared experiences can help in overcoming challenges. We have now better explained methods and improved discussion, to make this clear and to point the study limitations. 

6. PLOS authors have the option to publish the peer review history of their article (what does this mean?). If published, this will include your full peer review and any attached files.

Do you want your identity to be public for this peer review? For information about this choice, including consent withdrawal, please see our Privacy Policy.

Reviewer #1: No Reviewer #2: No

---

## [Decision Letter · Decision Letter 1]

23 Apr 2021

PONE-D-20-29941R1

Facing the COVID-19 Pandemic inside maternities in Brazil: challenges towards adequate care in the REBRACO initiative

PLOS ONE

Dear Dr. Costa,

Thank you for submitting your manuscript to PLOS ONE. After careful consideration, we feel that it has merit but does not fully meet PLOS ONE’s publication criteria as it currently stands. Therefore, we invite you to submit a revised version of the manuscript that addresses the points raised during the review process.

One of the 2 referees who revised the paper suggested “REJECTION”, based on the fact that “the paper is primarily descriptive without any comparison to another approach to maternity care during the pandemic. Without an historical or actual comparison to assess whether the changes/programs instituted in the participating centers had any effect on clinical outcomes, we cannot determine whether such changes provided any benefit that could be used now or in the future.  In addition, the paper still requires revision into proper scientific English”and further stated “I am still unclear as to the intent of study/paper.”. The second referee only suggested minor changes.

My opinion is that the paper can contribute by providing description of some changes underwent by the 16 hospitals of the Rebraco initiative in the first period of the pandemic, but it needs major changes to be suitable for publication.   I am here trying to help the paper authors by suggesting some changes that in my view could make the paper more suitable for publication.  

The paper currently lacks a good structure, and element of internal consistency, therefore and difficult to follow. I would suggest to check well consistency in between the study objective sentence> methods > results > discussion. These must be logically aligned. For example, the study objectives in the abstract (lines 4-5) does not mention the qualitative component. I suggest adding the study design in the title, this will facilitate readingSuggestions to improve the abstract_  here, beside revising the sentence on the study objectives,  you need to add key elements currently lacking, such as: 1) study time period ; 2) sample size; 3) more quantitative data (eg data available in Table 2 and 3); 3) avoid in the abstract vague wording such as “Changes in infrastructure and staff were reported” (line 18) or “Training session were less common” (line 20), these sentence adds very little since they are very unspecific and lack quantitative data; 4) be consistent with your study methods, so report key findings both quantitative and qualitative component; 5) the conclusion sentence need to the refer to the study findings, please see other papers as example; a typical sentence would be “study findings suggest that hospital of the REBRACO initiatives underwent major changes…; fear etc (qualitative component). Please note that the abstract is one of the key section of the paper, therefore if you wish to resubmit please consider revising it carefully.In order to improve the structure of the method section, I would suggest to add in the method section a paragraph for the quantitative component and one for the qualitative component. Both paragraphs would benefit of subparagraphs were all the following key element are clearly described: i) Population and setting, ii) data collection tools and methods, iii) data analysis.Please check carefully what belongs to each paragraph. For example, the current sentence and the beginning of the results section (lines 172-177) seems to be to pertain to the “study population” in the method section. It’s not clear if the sentence on the sample in lines 127-135 pertains to the quantitative or to the qualitative component.  The paper currently is very descriptive, often vague and/or not concise. A scientific paper needs to be concise and specific. Please if wishing to resubmit  revise text in order to be concise and specific. I suggest avoid in the paper text and in the abstract vague wording such as “data on several topics such as .. were collected” (line 102) and rather, make explicit exactly what is the list of variables which were collected (this could be done in a Supp Table). Grouping them consistently in methods and results (eg Infrastructures vs Human resources, as already done in Table 2) may also be helpful. Counting them is also helpful; for example, you may say in the method section that you collect data on XX key variable, covering the following domains: infrastructures (XX variables); human resources (XX variables); etc. This would be more concise and more specific that the current text.Consistency between the method section and the results section in terms of variable reported need to be ensured. Also, consistency in the discussion on key findings of both components need to be there.Be careful in wording. For example, revise sentences such as “the observational mixed-method component” (line 102). Your study design is mixed methods, and in it you have a quantitative and a qualitative component.Avoid in the introduction section sentences which rapidly become outdated, such as data on number of people affected by COVID (lines 33-36 and 36-38). Either delete them of make explicit that they refer to your study period.  The discussion section is quite long. I would suggest to cut it down, delete redundancies and flag out key concepts. Typically, a discussion sections include the following paragraphs: 1) key findings/ what this study adds (very concise, not a repetition of results section), 2) interpretation when comparing to existing literature; 3) study limitations; 4) Recommendations for research 5) recommendations for policy makers    You can find other inputs on how to improve your paper in the guideline of reporting for both observational cross-sectional studies (STROBE) and qualitative studies (see EQUATOR website). I suggest you bring major changes to your paper by following these guidelines. Please provide the 2 relevant checklists filled in.    

We look forward to receiving your revised manuscript.

Kind regards,

Marzia Lazzerini, PhD

Academic Editor

PLOS ONE

Additional Editor Comments (if provided):

One of the 2 referees who revised the paper suggested “REJECTION”, based on the fact that “the paper is primarily descriptive without any comparison to another approach to maternity care during the pandemic. Without an historical or actual comparison to assess whether the changes/programs instituted in the participating centers had any effect on clinical outcomes, we cannot determine whether such changes provided any benefit that could be used now or in the future. In addition, the paper still requires revision into proper scientific English”and further stated “I am still unclear as to the intent of study/paper.”. The second referee only suggested minor changes.

My opinion is that the paper can contribute by providing description of some changes underwent by the 16 hospitals of the Rebraco initiative in the first period of the pandemic, but it needs major changes to be suitable for publication. I am here trying to help the paper authors by suggesting some changes that in my view could make the paper more suitable for publication.

1) The paper currently lacks a good structure, and element of internal consistency, therefore and difficult to follow. I would suggest to check well consistency in between the study objective sentence> methods > results > discussion. These must be logically aligned. For example, the study objectives in the abstract (lines 4-5) does not mention the qualitative component.

2) I suggest adding the study design in the title, this will facilitate reading

3) Suggestions to improve the abstract_ here, beside revising the sentence on the study objectives, you need to add key elements currently lacking, such as: 1) study time period ; 2) sample size; 3) more quantitative data (eg data available in Table 2 and 3); 3) avoid in the abstract vague wording such as “Changes in infrastructure and staff were reported” (line 18) or “Training session were less common” (line 20), these sentence adds very little since they are very unspecific and lack quantitative data; 4) be consistent with your study methods, so report key findings both quantitative and qualitative component; 5) the conclusion sentence need to the refer to the study findings, please see other papers as example; a typical sentence would be “study findings suggest that hospital of the REBRACO initiatives underwent major changes…; fear etc (qualitative component). Please note that the abstract is one of the key section of the paper, therefore if you wish to resubmit please consider revising it carefully.

4) In order to improve the structure of the method section, I would suggest to add in the method section a paragraph for the quantitative component and one for the qualitative component. Both paragraphs would benefit of subparagraphs were all the following key element are clearly described: i) Population and setting, ii) data collection tools and methods, iii) data analysis.

5) Please check carefully what belongs to each paragraph. For example, the current sentence and the beginning of the results section (lines 172-177) seems to be to pertain to the “study population” in the method section. It’s not clear if the sentence on the sample in lines 127-135 pertains to the quantitative or to the qualitative component.

6) The paper currently is very descriptive, often vague and/or not concise. A scientific paper needs to be concise and specific. Please if wishing to resubmit revise text in order to be concise and specific. I suggest avoid in the paper text and in the abstract vague wording such as “data on several topics such as .. were collected” (line 102) and rather, make explicit exactly what is the list of variables which were collected (this could be done in a Supp Table). Grouping them consistently in methods and results (eg Infrastructures vs Human resources, as already done in Table 2) may also be helpful. Counting them is also helpful; for example, you may say in the method section that you collect data on XX key variable, covering the following domains: infrastructures (XX variables); human resources (XX variables); etc. This would be more concise and more specific that the current text.

7) Consistency between the method section and the results section in terms of variable reported need to be ensured. Also, consistency in the discussion on key findings of both components need to be there.

8) Be careful in wording. For example, revise sentences such as “the observational mixed-method component” (line 102). Your study design is mixed methods, and in it you have a quantitative and a qualitative component.

9) Avoid in the introduction section sentences which rapidly become outdated, such as data on number of people affected by COVID (lines 33-36 and 36-38). Either delete them of make explicit that they refer to your study period.

10) The discussion section is quite long. I would suggest to cut it down, delete redundancies and flag out key concepts. Typically, a discussion sections include the following paragraphs: 1) key findings/ what this study adds (very concise, not a repetition of results section), 2) interpretation when comparing to existing literature; 3) study limitations; 4) Recommendations for research 5) recommendations for policy makers

11) You can find other inputs on how to improve your paper in the guideline of reporting for both observational cross-sectional studies (STROBE) and qualitative studies (see EQUATOR website). I suggest you bring major changes to your paper by following these guidelines. Please provide the 2 relevant checklists filled in.

Reviewers' comments:

Reviewer's Responses to Questions

**Comments to the Author**

1. If the authors have adequately addressed your comments raised in a previous round of review and you feel that this manuscript is now acceptable for publication, you may indicate that here to bypass the “Comments to the Author” section, enter your conflict of interest statement in the “Confidential to Editor” section, and submit your "Accept" recommendation.

Reviewer #2: (No Response)

Reviewer #3: (No Response)

2. Is the manuscript technically sound, and do the data support the conclusions?

Reviewer #2: Partly

Reviewer #3: Yes

3. Has the statistical analysis been performed appropriately and rigorously? 

Reviewer #2: Yes

Reviewer #3: Yes

4. Have the authors made all data underlying the findings in their manuscript fully available?

Reviewer #2: Yes

Reviewer #3: Yes

5. Is the manuscript presented in an intelligible fashion and written in standard English?

Reviewer #2: No

Reviewer #3: No

6. Review Comments to the Author

Reviewer #2: I thank the authors for their revised manuscript. However, the paper is still primarily a descriptive manuscript without any comparison to another approach to maternity care during the pandemic. Without an historical or actual comparison to assess whether the changes/programs instituted in the participating centers had any effect on clinical outcomes, we cannot determine whether such changes provided any benefit that could be used now or in the future. In addition, the paper still requires revision into proper scientific English.

Reviewer #3: As I was not one of the initial reviewers, i have reviewed both the track changes version of the revised manuscript and the author's responses to prior reviewer comments. Generally I feel the manuscript is adequate in its present version, and look forward to seeing the platform developed through this collaboration used for more substantive contributions in perinatal epidemiology and health services research in the future.

Two minor comments. First, in the tables, be consistent in reporting percent values - these should be to one decimal, and include the trailing zero even when the value is zero (e.g. report 75.0% rather than 75%).

And second, while the manuscript is generally written in good English, a number of sentences are not, throughout the manuscript and especially in the abstract. Therefore I recommend, before finalization, the entire manuscript be reviewed by a scientist with English as a first language.

7. PLOS authors have the option to publish the peer review history of their article (what does this mean?). If published, this will include your full peer review and any attached files.

Reviewer #2: No

Reviewer #3: **Yes: **Russell S. Kirby, College of Public Health, University of South Florida, Tampa, Florida, USA

---

## [Author Response · Author response to Decision Letter 1]

26 Apr 2021

Point-by-Point Responses

Dear editor,

We acknowledge the detailed evaluation of our study.

Bellow, you will find a point-by-point response answering reviewer’s comments. All changes are highlighted (in yellow) in the manuscript.

Sincerely,

Maria Laura Costa, MD, PhD

Reviewer 1-

One of the 2 referees who revised the paper suggested “REJECTION”, based on the fact that “the paper is primarily descriptive without any comparison to another approach to maternity care during the pandemic. Without an historical or actual comparison to assess whether the changes/programs instituted in the participating centers had any effect on clinical outcomes, we cannot determine whether such changes provided any benefit that could be used now or in the future. In addition, the paper still requires revision into proper scientific English”and further stated “I am still unclear as to the intent of study/paper.”

RESPONSE:

We agree that the paper is mostly descriptive and that there is no comparison of time periods- this happened because it was submitted initially in September/2020, with data up to July/2021, during the first wave of the pandemic in Brazil. The aim of the study was to report the challenges in obstetrics and also in under resourced settings, with a quantitative and qualitative approach. The changes described are fundamental to ensure proper healthcare during the pandemic and provide adequate and safe environment both for patients and health workers and represented the prompt response of institutions in a timeframe of much uncertainty, lack of PPE (personal protective equipment) in many settings, need for extra ICU beds and resizing of emergency rooms and medical wards. 

At the time of submission, we already pointed towards awareness regarding maternal deaths due to COVID-19 in Brazil and unfortunately this concern, reported in this manuscript with 15 maternal deaths in a 5 months period was further evidenced. National data report a total of 5575 cases of severe COVID-19 disease among pregnant and postpartum women in 2020, with 386 maternal deaths (6.9%) (https://observatorioobstetrico.shinyapps.io/covid_gesta_puerp_br/).

Reviewer 2- 

The second referee only suggested minor changes.

RESPONSE: Thank you. Such comments were not reported.

Editor´s suggestions and comments:

My opinion is that the paper can contribute by providing description of some changes underwent by the 16 hospitals of the Rebraco initiative in the first period of the pandemic, but it needs major changes to be suitable for publication. I am here trying to help the paper authors by suggesting some changes that in my view could make the paper more suitable for publication. 

RESPONSE: Thank you. We acknowledge all suggestions and have worked to address them and improve the original submission. 

1. The paper currently lacks a good structure, and element of internal consistency, therefore and difficult to follow. I would suggest to check well consistency in between the study objective sentence> methods > results > discussion. These must be logically aligned. For example, the study objectives in the abstract (lines 4-5) does not mention the qualitative component. 

RESPONSE:

Thank you. We have made significant changes to address consistency throughout the manuscript. The adjusted sentences are highlighted. The aim of the study in the abstract was corrected:

“This study aims to describe the timeframe, implemented strategies and perspectives of health managers on the challenges to face the pandemic in 16 different maternity hospitals that comprise a multicenter study in Brazil, called REBRACO (Brazilian network of COVID-19 during pregnancy).” 

2. I suggest adding the study design in the title, this will facilitate reading 

RESPONSE: 

The title was changed according to the suggestion, to include the study design

“Facing the COVID-19 Pandemic inside maternities in Brazil: a mixed-method study within the REBRACO initiative”

3. Suggestions to improve the abstract_ here, beside revising the sentence on the study objectives, you need to add key elements currently lacking, such as: 1) study time period ; 2) sample size; 3) more quantitative data (eg data available in Table 2 and 3); 3) avoid in the abstract vague wording such as “Changes in infrastructure and staff were reported” (line 18) or “Training session were less common” (line 20), these sentence adds very little since they are very unspecific and lack quantitative data; 4) be consistent with your study methods, so report key findings both quantitative and qualitative component; 5) the conclusion sentence need to the refer to the study findings, please see other papers as example; a typical sentence would be “study findings suggest that hospital of the REBRACO initiatives underwent major changes…; fear etc (qualitative component). Please note that the abstract is one of the key section of the paper, therefore if you wish to resubmit please consider revising it carefully. 

RESPONSE:

Thank you. All information was included/adjusted accordingly and detailed bellow in italic. 

1) study time period: from January to July/2020 (line 11);

2) sample size: for the qualitative study: “A qualitative study by purposeful and saturation sampling was undertaken with healthcare managers”; for the quantitative study, we considered all cases of COVID-19 reported by each participating center and their information on total number of deliveries, livebirth, maternal deaths. 

3) more quantitative data (eg data available in Table 2 and 3. avoid in the abstract vague wording such as “Changes in infrastructure and staff were reported” (line 18) or “Training session were less common” (line 20), these sentence adds very little since they are very unspecific and lack quantitative data;

The required information was added: All maternities performed relocation of beds designated to labor ward, most (75%) acquired mechanical ventilators, only the minority (25%) installed new negative air pressure rooms. Considering human resources, around 40% hired extra health professionals and increased weekly workload and the majority (68.7%) also suspended annual leaves. Only one center implemented universal screening for childbirth. Qualitative results showed that main challenges experienced were related to the fear of the virus, concerns about reliability of evidence, lack of resources and clear need for mental health support among health professionals.

4) be consistent with your study methods, so report key findings both quantitative and qualitative component- Improved as reported above.

5) the conclusion sentence need to the refer to the study findings, please see other papers as example; a typical sentence would be “study findings suggest that hospital of the REBRACO initiatives underwent major changes…; fear etc (qualitative component). Please note that the abstract is one of the key section of the paper, therefore if you wish to resubmit please consider revising it carefully.

Thank you. Conclusion now refers to main study findings:

“Study findings suggest that maternities of the REBRACO initiative underwent major changes in facing the pandemic, mostly resizing emergency rooms, relocating beds and adding extra ICU units and new ventilators. On human resources, most centers had to employ increased weekly workloads and listening to health managers, main concerns included fear of the virus and lack of resources. In the first months of the pandemic, there were 15 maternal deaths due to COVID-19 in the reported centers.”

4. In order to improve the structure of the method section, I would suggest to add in the method section a paragraph for the quantitative component and one for the qualitative component. Both paragraphs would benefit of subparagraphs were all the following key element are clearly described: i) Population and setting, ii) data collection tools and methods, iii) data analysis. 

RESPONSE:

Thank you. We have now organized methods in two subgroups (quantitative component and qualitative component) with the required information.

5. Please check carefully what belongs to each paragraph. For example, the current sentence and the beginning of the results section (lines 172-177) seems to be to pertain to the “study population” in the method section. It’s not clear if the sentence on the sample in lines 127-135 pertains to the quantitative or to the qualitative component. 

RESPONSE:

Thank you. We agree, the first sentence in results belongs to methods and was excluded. 

The information on informed consent for each component of the study is now more clear in the methods, with the previous suggestion of better defining quantitative and qualitative approaches (with sub-headings added). 

6. The paper currently is very descriptive, often vague and/or not concise. A scientific paper needs to be concise and specific. Please if wishing to resubmit revise text in order to be concise and specific. I suggest avoid in the paper text and in the abstract vague wording such as “data on several topics such as .. were collected” (line 102) and rather, make explicit exactly what is the list of variables which were collected (this could be done in a Supp Table). Grouping them consistently in methods and results (eg Infrastructures vs Human resources, as already done in Table 2) may also be helpful. Counting them is also helpful; for example, you may say in the method section that you collect data on XX key variable, covering the following domains: infrastructures (XX variables); human resources (XX variables); etc. This would be more concise and more specific that the current text. 

RESPONSE: 

Thank you. We have included all suggestions in the text and added the Supporting Box bellow with the data requested”. The paragraph now is:

“For the quantitative component, we collected data on the infrastructure/equipment of the units, maternal and perinatal health indicators, characteristics of service provision and modifications on staff and human resources. Detailed material can be found in the Supporting Information (S1)”.

Supporting information-S1

S1: Variables requested for the characterization of the included REBRACO centers. List of variables which were collected through online forms

Infrastructure/Equipment

Number of beds designated for labor ward (per month, from March to August)

Number of beds designated for obstetric medical ward (per month, from March to August)

Number of beds designated for intensive care (per month, from March to August)

Availability of the tests to confirm the infection by SARS-CoV 2 and number of tested women (per month, from March to August)

Implementation of modifications in the maternity´s infra-structure (resizing wards, relocation of beds, acquiring new equipments, such as ventilators)

Availability of PPE (Personal Protective Equipment)

Maternal and perinatal health indicators

Total number of attendances at the obstetric emergency care unit (per month, from March to August)

Number of attendances at the obstetric emergency care unit (per month, from March to August) for pregnant and postpartum women with suspected COVID-19

Number of new confirmed cases of COVID-19 in pregnant and postpartum women (per month, from March to August)

Criteria for suspected cases of COVID-19

Number of hospitalizations of pregnant and postpartum women in the unit (per month, from March to August)

Number of hospitalizations due to COVID-19 in pregnant and postpartum women in the unit (per month, from March to August) (suspected or confirmed)

Number of new cases of severe acute respiratory syndrome (SARS) in pregnant/postpartum women, in the unit (per month, from March to August)

Number of new cases of severe acute respiratory syndrome (SARS) caused by COVID-19 in pregnant/postpartum women, in the unit (per month, from March to August)

Number of maternal deaths due to all causes in the unit (per month, from March to August)

Number of maternal deaths due to COVID-19 unit (per month, from March to August)

Number of live births in you unit (per month, from March to August)

Cesarean section rate (%; elective or intrapartum cesarean) in the unit (per month, from March to August)

Number of live births in the municipality unit (per month, from March to August)

Number of stillbirths in the unit (per month, from March to August)

Number of stillbirths in the municipality unit (per month, from March to August)

Other characteristics of service provision

Referral center for cases suspected/confirmed for COVID-19 (y/n)

Rules on the presence of companions during childbirth (always, often, sometimes, rarely or never; per month, from March to August)

How long on average was the turnover to obtain test results for COVID-19? (in days; per month, from March to August)

Gynecological and obstetrical services according to the usual standard in the unit (suspended, reduction, no change, not applicable)

Staff/Human Resources

Presence of Ob&Gyn medical residents (y/n)

Number of professionals that received sick leave due to COVID-19 in the unit (per month, from March to August)

Members of the COVID-19 Emergency Action Committee (EAC) - (according to type of health professionals) and date of EAC implementation (from January to August)

Modification of work schedule of health professionals (Hiring new professionals, increase in weekly workload, suspension of annual leaves)

Content of the training program for COVID-19 

7. Consistency between the method section and the results section in terms of variable reported need to be ensured. Also, consistency in the discussion on key findings of both components need to be there. 

RESPONSE: 

Thank you. We have revised the manuscript for consistency, included data that were not reported in tables (table 1 was revised) and better reported results and organized the discussion. 

This paragraph is now referred to table 1, for example:

“All maternities established a local Covid-19 Emergency Action Committee (EAC) and implemented a local protocol for contingency and management of COVID-19 in pregnancy. The EAC was usually comprised of a general practitioner or internal medicine (73% of the centers, n=11), Obstetrics and Gynecology specialist (75%, n=12), intensive care specialist (68.7%, n=11), nurse (75%, n=12), member of the Infection Prevention and Control Committee (75%, n=12), administrative assistant (68.7%, n=11), and, less often, of a physiotherapist (31.2%, n=5) (Table 1). All the 16 maternities had access to diagnostic laboratory testing. However, only 6 maternities (37.5%) had resources to test any suspected case of COVID-19.”

8. Be careful in wording. For example, revise sentences such as “the observational mixed-method component” (line 102). Your study design is mixed methods, and in it you have a quantitative and a qualitative component. 

RESPONSE: 

Thank you. We have corrected the wording throughout the manuscript. 

9. Avoid in the introduction section sentences which rapidly become outdated, such as data on number of people affected by COVID (lines 33-36 and 36-38). Either delete them of make explicit that they refer to your study period. 

RESPONSE:

We agree and have modified the paragraph to avoid too many numbers and include only data consistent with the considered study period, in Brazil. 

“Coronavirus disease 2019 (COVID-19) has threatened the world, since March 2020, when the pandemic was officially declared by the WHO (World Health Organization) [1]. Some countries were especially affected and faced individual challenges towards viral dissemination. Brazil was certainly one of those, recognized as a pandemic hotspot with more than 4.1 million people infected and over 126,000 deaths, in the first 6 months of the pandemic”

10. The discussion section is quite long. I would suggest to cut it down, delete redundancies and flag out key concepts. Typically, a discussion sections include the following paragraphs: 1) key findings/ what this study adds (very concise, not a repetition of results section), 2) interpretation when comparing to existing literature; 3) study limitations; 4) Recommendations for research 5) recommendations for policy makers 

RESPONSE: 

Thank you. We have reviewed discussion, excluded redundancies (three paragraphs) and better organized ideas, according to the suggestions above. 

Now, the first paragraph addresses the key findings:

“The 16 maternities of the REBRACO initiative were mostly university hospitals, which underwent major changes and adaptations in facing the first few months of the pandemic, resizing emergency rooms, relocating beds and adding extra ICU units, with acquisition of ventilators. On human resources, most centers had to employ increased weekly workloads, deal with medical leave due to COVID infections, and hire extra health professionals. Listening to health managers, main concerns included fear of the virus and lack of resources. There was a clear limitation in the capacity of testing during the considered study period, less than half of the centers had resources to test any suspected case of COVID-19.”

 We again, acknowledge all suggestions and hope that the revised version of the manuscript is suitable for publication. 

Best regards,

Maria Laura Costa

---

## [Editor Report · Decision Letter 2]

18 May 2021

PONE-D-20-29941R2

Facing the COVID-19 Pandemic inside maternities in Brazil: a mixed-method study within the REBRACO initiative

PLOS ONE

Dear Dr. Costa,

Thank you for submitting your manuscript to PLOS ONE. After careful consideration, we feel that it has merit but does not fully meet PLOS ONE’s publication criteria as it currently stands. Therefore, we invite you to submit a revised version of the manuscript that addresses the points raised during the review process.

The paper has improved a lot, but need further improvement. Again, I will give here some suggestions on how to improve it:

ABSTRACT

Line 4: is the “the timeframe” really one of the indicators? Where is it in results? Otherwise delete from abstract and also from the text 

Line 7-10. Find a way to be more concise  on quantitative indicators.

Line 12 versus line 4: either use the word “perspective” (line 4) or “local challenges” (line 12). The 2 words do not have the same meaning. Or should it be perspectives on local challenges?

Line 14: delete “Rebraco centers respond for a total of 4000 deliveries/months “ or move it to introduction

Line 21 “while most only tested” : once again I advise to avoid using unspecific words such as most

Conclusion: line 26-30 are just a repetition of study findings, delete these and add in conclusion recommendations for policy makers and for further research.  

INTRO

Line 42-45   are not relevant to your research question. Please revise to make clear what are the knowledge gaps related to HS and why is your research question relevant

Line 59: you state that the REBRACO collect qualitative data from women experiences, but there is no further mention on this in the paper

Line 63-65. Add that you also collected perspectives of HW

METHODS

Line 67-69 and 73-75 are a repetition of lines 54-57. I suggest to place the background information on REBRACO in the Introduction. Here in methods add details on study population and settings- Move here lines 165-169

Line 76-78 and 99-101 repeated the study objectives again

Line 102-132 and 133-162 These sections still lack some structure. Please first describe what data you collected, then how, and then how the analysis performed. Separate paragraphs so that they are clearly identified

Table 1. there are only few quantitative indicators on your research question in this table, consider adding in tables data that now are in the text

CONCLUSION

Still very long, please reduce length

. 

We look forward to receiving your revised manuscript.

Kind regards,

Marzia Lazzerini, PhD

Academic Editor

PLOS ONE

Journal Requirements:

Additional Editor Comments (if provided):

The paper has improved a lot, but need further improvement. Again, I will give here some suggestions on how to improve it:

ABSTRACT

Line 4: is the “the timeframe” really one of the indicators? Where is it in results? Otherwise delete from abstract and also from the text

Line 7-10. Find a way to be more concise on quantitative indicators.

Line 12 versus line 4: either use the word “perspective” (line 4) or “local challenges” (line 12). The 2 words do not have the same meaning. Or should it be perspectives on local challenges?

Line 14: delete “Rebraco centers respond for a total of 4000 deliveries/months “ or move it to introduction

Line 21 “while most only tested” : once again I advise to avoid using unspecific words such as most

Conclusion: line 26-30 are just a repetition of study findings, delete these and add in conclusion recommendations for policy makers and for further research.

INTRO

Line 42-45 are not relevant to your research question. Please revise to make clear what are the knowledge gaps related to HS and why is your research question relevant

Line 59: you state that the REBRACO collect qualitative data from women experiences, but there is no further mention on this in the paper

Line 63-65. Add that you also collected perspectives of HW

METHODS

Line 67-69 and 73-75 are a repetition of lines 54-57. I suggest to place the background information on REBRACO in the Introduction. Here in methods add details on study population and settings- Move here lines 165-169

Line 76-78 and 99-101 repeated the study objectives again

Line 102-132 and 133-162 These sections still lack some structure. Please first describe what data you collected, then how, and then how the analysis performed. Separate paragraphs so that they are clearly identified

Table 1. there are only few quantitative indicators on your research question in this table, consider adding in tables data that now are in the text

CONCLUSION

Still very long, please reduce length

---

## [Author Response · Author response to Decision Letter 2]

20 May 2021

Dear editor,

We again acknowledge the detailed evaluation of our study. We agree that all previous suggestions and comments have greatly improved the manuscript: “Facing the COVID-19 Pandemic inside maternities in Brazil: a mixed-method study within the REBRACO initiative”. 

Bellow, you will find a point-by-point response answering reviewer’s comments. All changes are highlighted (in yellow) in the manuscript.

Sincerely,

Maria Laura Costa, MD, PhD

Reviewer -

The paper has improved a lot, but need further improvement. Again, I will give here some suggestions on how to improve it:

ABSTRACT

Line 4: is the “the timeframe” really one of the indicators? Where is it in results? Otherwise delete from abstract and also from the text 

RESPONSE: We agree, the term was confusing. We have deleted. The aim now is: 

“This study aims to describe the strategies implemented and the perspectives of health managers on the challenges to face the pandemic in 16 different maternity hospitals that comprise a multicenter study in Brazil, called REBRACO (Brazilian network of COVID-19 during pregnancy).

It was also deleted from the text. 

Line 7-10. Find a way to be more concise on quantitative indicators.

RESPONSE: Thank you. We have made it more concise. It now reads:

“Mixed-method study, with quantitative and qualitative approaches. Quantitative data on the infrastructure of the units, maternal and perinatal health indicators, modifications on staff and human resources, from January to July/2020. Also, information on total number of cases, and availability for COVID-19 testing. A qualitative study by purposeful and saturation sampling was undertaken with healthcare managers, to understand perspectives on local challenges in facing the pandemic.”

Line 12 versus line 4: either use the word “perspective” (line 4) or “local challenges” (line 12). The 2 words do not have the same meaning. Or should it be perspectives on local challenges?

RESPONSE: Thank you. We believe the most accurate would be “perspectives on local challenges”. We have changed the text. 

Line 14: delete “Rebraco centers respond for a total of 4000 deliveries/months “ or move it to introduction

RESPONSE: It was deleted. 

Line 21 “while most only tested” : once again I advise to avoid using unspecific words such as most

RESPONSE: Thank you. We have included the complete information:

“while around 60% of the centers only tested moderate/severe cases with hospital admission”

Conclusion: line 26-30 are just a repetition of study findings, delete these and add in conclusion recommendations for policy makers and for further research. 

 RESPONSE: Thank you. We have adjusted the conclusion in order to address main findings and recommendations for policy makers and for further research. 

“Study findings suggest that maternities of the REBRACO initiative underwent major changes in facing the pandemic, with limitations on testing, difficulties in infrastructure and human resources. Leadership, continuous training, implementation of evidence-based protocols and collaborative initiatives are key to transpose the fear of the virus and ascertain adequate healthcare inside maternities, especially in low and middle-income settings. Policy makers need to address the specificities in considering reproductive health and childbirth during the COVID-19 pandemic and prioritize research and timely testing availability.”

INTRO

Line 42-45 are not relevant to your research question. Please revise to make clear what are the knowledge gaps related to HS and why is your research question relevant

RESPONSE: Thank you, it really was not clear. The sentence needs to present the challenge that the country size, economic and political crises can add to a pandemic. We have changed it to:

“Brazil is an upper-middle-income country of continental size, with great social and economic disparities among different regions, and major political crisis. These characteristics enhance the challenges in facing a pandemic”. 

Line 59: you state that the REBRACO collect qualitative data from women experiences, but there is no further mention on this in the paper

RESPONSE: Thank you. We have adjusted the text to make sure that it is clear that we are describing the main study, and how the initiative was built, with the reference for the overall protocol (actually a Letter that describes the main objectives of the project, as a “call to action”). Therefore, in the current analysis we only included the qualitative data on health professionals. 

“Overall, the REBRACO initiative aimed at evaluating the clinical, epidemiological and laboratory aspects related to SARS-CoV-2 infection during pregnancy, besides a qualitative assessment of women and professionals experiencing such a situation, to identify maternal and perinatal outcomes and collect relevant information to provide quick responses and proper organization of health services to confront the COVID-19 pandemic [11]. 

As part of the main study, we aim to describe the strategies implemented and the perspectives of health managers on the challenges to face the pandemic in 16 different maternity hospitals …”

Line 63-65. Add that you also collected perspectives of HW

RESPONSE: we have added the information on HW perspectives.

METHODS

Line 67-69 and 73-75 are a repetition of lines 54-57. I suggest to place the background information on REBRACO in the Introduction. Here in methods add details on study population and settings- Move here lines 165-169

RESPONSE: Thank you. We have deleted repeated information. 

Line 76-78 and 99-101 repeated the study objectives again

RESPONSE: Repetition was deleted.

Line 102-132 and 133-162 These sections still lack some structure. Please first describe what data you collected, then how, and then how the analysis performed. Separate paragraphs so that they are clearly identified

RESPONSE: Thank you. We agree that information was not well organized. We have structured as suggested. 

“For the quantitative component, we collected data on the infrastructure/equipment of the units, maternal and perinatal health indicators, characteristics of service provision and modifications on staff and human resources. All information was retrieved through an electronic form developed by the coordinating center and sent by e-mail to the Principal Investigator of each center (Health Manager, responsible for the Obstetrical Emergency Action Committee - EAC - in each institution). The data collection form is provided as Supporting Information (S1) and includes data on: characteristics of the maternities, their response program, their training program and some indicators related to the COVID-19 pandemic such as the number of cases, number of maternal deaths, maternal mortality ratio due to COVID-19 in the period, the time interval between the implementation of the response program and the first suspected COVID-19 cases in each center and the characteristics of the tests available in the obstetric units.

All suspected cases of COVID-19 in the considered maternities were assessed for informed consent and their data was only included after such consent, with identity kept confidential. Considering biosafety reasons, the study has authorization, by the Institutional Review Board, for both- written informed consent and consent by phone to use information from medical records and sample collection. Mostly suspected cases with hospital admissions have written informed consent, cases that were evaluated and tested in the emergency room, with no needed admission were later consented by phone. The researcher read and explained the consent form and assured that women could withdraw from the study without any interference with planned medical care. All interviews were recorded on audio after consent from the participants, using the Microsoft Skype platform.“

Table 1. there are only few quantitative indicators on your research question in this table, consider adding in tables data that now are in the text

RESPONSE: We have adjusted the Table description in order to present clearly the results:

“Table 1. Characteristics of included maternities in the REBRACO study (n=16), participating members of the local Emergency Action Committees (EAC) and information on COVID-19 infection and some healthcare indicators.”

CONCLUSION

Still very long, please reduce length

RESPONSE: it was reduced. 

“Study findings suggest that maternities of the REBRACO initiative underwent major changes in facing the pandemic, with limitations on testing, difficulties in infrastructure and human resources. Leadership, continuous training, implementation of evidence-based protocols and collaborative initiatives are key to transpose the fear of the virus and ascertain adequate healthcare inside maternities, especially in low and middle-income settings. Policy makers need to address the specificities in considering reproductive health and childbirth during the COVID-19 pandemic and prioritize research and timely testing availability.”

We again, acknowledge all suggestions and hope that the revised version of the manuscript is suitable for publication. 

Best regards,

Maria Laura Costa

---

## [Editor Report · Decision Letter 3]

8 Jul 2021

Facing the COVID-19 Pandemic inside maternities in Brazil: a mixed-method study within the REBRACO initiative

PONE-D-20-29941R3

Dear Dr. Costa,

We’re pleased to inform you that your manuscript has been judged scientifically suitable for publication and will be formally accepted for publication once it meets all outstanding technical requirements.

Kind regards,

Marzia Lazzerini, PhD

Academic Editor

PLOS ONE

Additional Editor Comments (optional):

After the lasT revision I feel that the paper is suitable fopr publication if:

- yuo add when data were collected (ie,month, year)

- check is sentence on line 116-117 is pertinent to qunatittaive data collection ("All interviews were recorded on audio after consent 117 from the participants, using the Microsoft Skype platform" Explain how interviews were pertinent to quantittaive data)
---

## [Editor Report · Acceptance letter]

16 Jul 2021

PONE-D-20-29941R3 

Facing the COVID-19 Pandemic inside maternities in Brazil: a mixed-method study within the REBRACO initiative 

Dear Dr. Costa:

I'm pleased to inform you that your manuscript has been deemed suitable for publication in PLOS ONE. Congratulations! Your manuscript is now with our production department. 

Kind regards, 

on behalf of

Dr. Marzia Lazzerini 

Academic Editor

PLOS ONE